# Ultrafast piezocapacitive soft pressure sensors with over 10 kHz bandwidth via bonded microstructured interfaces

Yuan Zhang[1,7], Xiaomeng Zhou [2,7], Nian Zhang[3,7], Jiaqi Zhu[1], Ningning Bai[1], Xingyu Hou[1], Tao Sun[4], Gang Li[1], Lingyu Zhao[1], Yingchun Chen[5] ✉, Liu Wang [3,6] ✉ & Chuan Fei Guo [1] ✉

Flexible pressure sensors can convert mechanical stimuli to electrical signals to interact with the surroundings, mimicking the functionality of the human skins. Piezocapacitive pressure sensors, a class of most widely used devices for artificial skins, however, often suffer from slow response-relaxation speed (tens of milliseconds) and thus fail to detect dynamic stimuli or high-frequency vibrations. Here, we show that the contact-separation behavior of the electrode-dielectric interface is an energy dissipation process that substantially determines the response-relaxation time of the sensors. We thus reduce the response and relaxation time to ~0.04 ms using a bonded microstructured interface that effectively diminishes interfacial friction and energy dissipation. The high response-relaxation speed allows the sensor to detect vibrations over 10 kHz, which enables not only dynamic force detection, but also acoustic applications. This sensor also shows negligible hysteresis to precisely track dynamic stimuli. Our work opens a path that can substantially promote the response-relaxation speed of piezocapacitive pressure sensors into submillisecond range and extend their applications in acoustic range.

The perception of touch of the human skin is enabled by mechanoreceptors that respond to not only static forces (by slow adaptors) but also vibrational stimuli (by fast adaptors)[1]. Electronic skins or flexible pressure sensors are emerging devices that mimic the functionalities of the mechanoreceptors[2–5], which have been widely studied because of their potential applications in the fields of robot haptics[6–8], human-machine interfaces[9,10], intelligent wearables[11–13], and metaverse[14–16]. Many applications, such as texture recognition, sound recognition, and pressure/vibration detection, require sensors to respond to both static pressure and high-frequency vibrations up to thousands of hertz.

Piezocapacitive flexible pressure sensors are a class of the most widely studied sensing devices that can detect static pressure, but these devices perform insufficiently in responding to dynamic stimuli. While elastic elastomers can respond to mechanical stimuli in nanoseconds[17–19], conventional piezocapacitive flexible pressure sensors often exhibit a response-relaxation time on the level of tens of milliseconds, corresponding to a narrow frequency range up to tens of hertz. This low response-relaxation speed is primarily attributed to energy dissipation associated with viscoelastic materials and interfacial frictions. Soft dielectrics are typically viscoelastic materials that

¹Department of Materials Science and Engineering, Southern University of Science and Technology, Shenzhen 518055, China. ²CAS Key Laboratory of Human-Machine Intelligence-Synergy Systems, Shenzhen Institutes of Advanced Technology, Chinese Academy of Sciences, Shenzhen 518055, China. ³CAS Key Laboratory of Mechanical Behavior and Design of Materials, University of Science and Technology of China, Hefei 230000, China. ⁴Department of Computer Science and Engineering, Southern University of Science and Technology, Shenzhen 518055, China. ⁵Science and Technology Committee, Commercial Aircraft Corporation of China Ltd., Shanghai 200126, China. ⁶State Key Laboratory of Nonlinear Mechanics, Institute of Mechanics, Chinese Academy of Science, Beijing 100190, China. ⁷These authors contributed equally: Yuan Zhang, Xiaomeng Zhou, Nian Zhang. ✉e-mail: chenyingchun@comac.cc; wangliu05@ustc.edu.cn; guocf@sustech.edu.cn

dissipate energy during loading-unloading cycles. Such an energy loss gets more pronounced when softer materials are used for detecting subtle pressures[20,21]. Also, during the contact-separation process, the interfacial friction and adhesion between the electrode and dielectric further contribute to energy loss[22,23]. To improve the response-relaxation speed, a common strategy is to engineer the dielectric layer with microstructured surfaces[20]. This strategy works through two principles. First, the microstructures reduce the bulk viscoelasticity of the dielectric by storing more elastic energy in smaller deformations. Second, they reduce the contact area between the dielectric and electrode, thereby lowering energy dissipation due to interfacial friction and adhesion. However, despite the reduced energy dissipation by introducing microstructures, the response-relaxation time remains largely above 1 ms to date. This limitation seems to be unreconcilable as long as viscoelastic materials are used and interfacial gaps persist. Although there are a few very recent advances reporting sensors with a shorter response-relaxation time down to a few milliseconds[6,24–26], such sensors can still not be used to detect high-frequency vibrations of hundreds or thousands of hertz, and thus the application of the sensors to high-frequency or acoustic purpose is still unavailable.

In this work, we present a strategy for downscaling the response-relaxation time of flexible piezocapacitive pressure sensors to ~0.04 ms by seamlessly bonding a low-viscosity microstructured dielectric with the electrode. The dielectric is made by dispersing 2 wt.% carbon nanotubes (CNTs) within a polydimethylsiloxane (PDMS) matrix, which reduces the material viscosity and surface adhesion. Without interfacial gaps, the bonded microstructured interfaces substantially diminish the friction-induced energy dissipation. We show that our sensor can quickly respond to stimuli from steady pressures to high-frequency vibrations over 10 kHz. In addition, the sensor also exhibits a high frequency-resolution of 0.2 Hz at 1000 Hz, and negligible capacitance-pressure hysteresis. Such behaviors enable its applications for dynamic pressure detection including acoustic scenarios. We further designed an artificial ear system based on the sensor and used the system for sound detection. We expect that our sensor to be used in more applications that require the detection of both static pressure and high-vibrational stimuli, and the method of using a bonded interface to improve response-relaxation speed might be extended to other devices.

## Results

### Microstructured pressure sensors with a bonded interface

Conventional microstructured pressure sensors have a gap between the electrode and the dielectric (Fig. 1a), while our sensors have a bonded electrode-dielectric interface (Fig. 1b). To compare their different response-relaxation behavior, we fabricated two sensors. The top and bottom electrodes are made by dispersing 7 wt.% carbon nanotubes (CNTs) in polydimethylsiloxane (PDMS), while the middle dielectric layer is made of PDMS with 2 wt.% CNT fillers. Microstructured dielectric layers were fabricated by molding a 3D-printed template of microcones (Supplementary Fig. 1). The bonded interface is further realized by immersing electrodes and dielectric in trichloromethane solvent with uncured PDMS networks, followed by a curing process to form topological interlinks (Supplementary Fig. 2)[27].

To elucidate the reduced energy dissipation of bonded interfaces, we performed finite element analysis (FEA) in which two sensors are compressed to 100 kPa and then recovered (Fig. 1c and Supplementary Fig. 3). The bottom area of the microcone and the increased electrode-dielectric contact area is denoted as $D$ and $\Delta A$, respectively. For the non-bonded sensor with a gap, $\Delta A$ increases quickly upon loading, as shown in Fig. 1d, and the stress concentrates at contacted regions (Fig. 1c). The corresponding unloading process is shown in Supplementary Fig. 3. As a result, the friction contributes to substantial energy loss (denoted as $W_{\text{loss}}$) that is higher than the maximum elastic energy (denoted as $W_{\text{elastic}}$) during loading–unloading process

(Fig. 1e). On the contrary, the bonded pressure sensor only shortens under compression with negligibly smaller $\Delta A$ than that of the non-bonded counterpart (Fig. 1d). Therefore, the energy loss is significantly reduced (Fig. 1e), leading to an elevated response-relaxation time of the bonded sensor.

The bonded pressure sensor exhibits a variation of capacitance $\Delta C$ to a broad pressure range of 0−350 kPa (Fig. 1f) with exceptionally rapid response-relaxation speed. Figure 1g shows that both the response time and the relaxation time are 0.04 ms, which is tested by a customized circuit board (Supplementary Figs. 4 and 5, Supplementary Text 1). The maximum frequency that a sensor can detect is determined as: $f_{\text{max}} = 1/(t_{\text{res}} + t_{\text{rel}})$, where $t_{\text{res}}$ and $t_{\text{rel}}$ represent response time and relaxation time, respectively. Existing piezo-capacitive sensors mostly exhibit a response-relaxation time of tens of milliseconds, and the corresponding $f_{\text{max}}$ is on the order of tens of hertz. Here, the total response-relaxation time of 0.08 ms allows the sensor to respond to high frequencies up to 12,500 Hz. By contrast, the sensor with a non-bonded interface exhibits a much longer response time of 4.76 ms and a relaxation time of 6.55 ms, which allows the sensor to respond to a frequency limit of only 88 Hz. Moreover, our sensor outperforms existing capacitive pressure sensors in both response and relaxation speed (Fig. 1h) by one to two orders of magnitude compared with existing capacitive pressure sensors with a non-bonded interface[3,7,24,28–42]. Note that the response time and relaxation time of piezocapacitive sensors are correlated and exhibit close values (the dashed line in Fig. 1h). In general, the ratio of relaxation time to response time is usually greater than or close to 1. In addition to the wide frequency bandwidth up to 12,500 Hz, our sensor exhibits a low detection limit of 0.007 Pa, which allows the sensor to detect tiny stimuli such as sound. By contrast, existing sensors can hardly balance both a low detection limit and a wide frequency bandwidth (shown as in Fig. 1i)[28,35,43–49].

### Effect of microcones on response-relaxation time

The bonded microcones play a key role in reducing the energy dissipation during the contact-separation process, thus increasing the response-relaxation speed. The effect of microcone structure on the response-relaxation time is further investigated by FEA. The structure of a single microcone in 2D configuration can be described by three parameters: height $H$, top surface $A_0$ (i.e., initial contact area), and bottom surface $D$ (Fig. 2a). Twenty different microcones are investigated by varying $A_0/D = 0.2, 0.4, 0.6, 0.8, 1$ and $H/D = 0.5, 1, 1.5, 2$. With a fixed $A_0/D = 0.4$, we first study the effect of height $H$ on the energy dissipation by varying $H/D$. Results in Fig. 2b show that sharper microcones (i.e., larger $H/D$) show less energy loss. However, sharp microcones may also undergo buckling instability that undermines the mechanical stability of the sensor. For a tapered column with clamped top/bottom surfaces, the critical buckling force can be calculated by $\frac{4\pi^2 E I_D}{H^2}\frac{A_0^2}{D^2}$, where $E$ is the Young's modulus and $I_D = \frac{\pi D^4}{64}$[50]. The critical buckling pressure (denoted as $P_c$) of 20 microcones are summarized in Fig. 2c. Considering that the maximum applied pressure should be greater than 300 kPa, microcones with a small $A_0/D$ and a large $H/D$ should not be adopted (marked with blue in Fig. 2c).

Second, we analyze how initial contact area $A_0$ affects the energy dissipation by setting $H/D = 1$ and varying $A_0/D$. The normalized energy loss in Fig. 2d suggests that larger $A_0/D$ yields a lower energy loss. This conclusion is expected because stout microcones (i.e., larger $A_0/D$) create a smaller contact area under the same pressure while preserving a larger elastic energy $W_{\text{elastic}}$. However, we may not rush to conclude that microcone with $A_0/D = 1$ (i.e., a cylindrical micropillar) is the best by far, because the flat dielectric is notorious for its low sensitivity[20], i.e., the slope of $\Delta C/C_0$ versus pressure curve. As also revealed by our models in Fig. 2e, we show that the sensitivity gradually drops when $A_0/D$ increases (Supplementary Text 2), leading to an undermined detection range due to the saturated signals, as validated by

experiments (dotted data in Fig. 2e). Therefore, the tradeoff between response-relaxation speed and sensitivity should also be considered. By summarizing the normalized energy loss of 20 microcones in Fig. 2f with background colors indicating other metrics, we show that microcones with moderate values of $(A_0/D, H/D)$ = (0.4, 1), (0.6, 1) and (0.6, 1.5) can simultaneously achieve low energy dissipation, high sensitivity, and high mechanical stability. Note that it is difficult to fabricate bonded microcones with perfect geometries, and microcones used in this work have close values of $(A_0/D, H/D) \approx (0.4, 1)$ (see SEM in Fig. 1b). In addition to the structure of microcones, the bonding ratio of microcone arrays also affect the response-relaxation time of the sensor. In this regard, we compare the performance of six sensors with bonding ratios of 0, 20, 40, 60, 80, and 100% (Fig. 2g). Results in Fig. 2h manifest that the 100% bonded ratio provides the lowest

response-relaxation time of ~0.04 ms, which is higher than previous sensors without fully bonded interfaces (Supplementary Fig. 6)[27].

## Effect of materials on response-relaxation time

Viscoelastic materials are known for their energy dissipation during loading-unloading cycles. To reduce the viscosity of the dielectric, we employ a highly crosslinked PDMS matrix with a low base-to-curing agent ratio of 5:1, into which CNT fillers are further introduced. The choice of CNTs is due to their stable mechanical and superior electrical properties. Notably, the PDMS-CNTs composite's relative permittivity is significantly higher than that of pure PDMS, thereby enhancing the capacitance signal[51]. As shown in Fig. 3a, the 5:1 PDMS has a lower ratio of loss modulus to storage modulus ($E''/E'$) compared to the 15:1 PDMS. By further doping CNTs, the PDMS-CNTs composite exhibits even

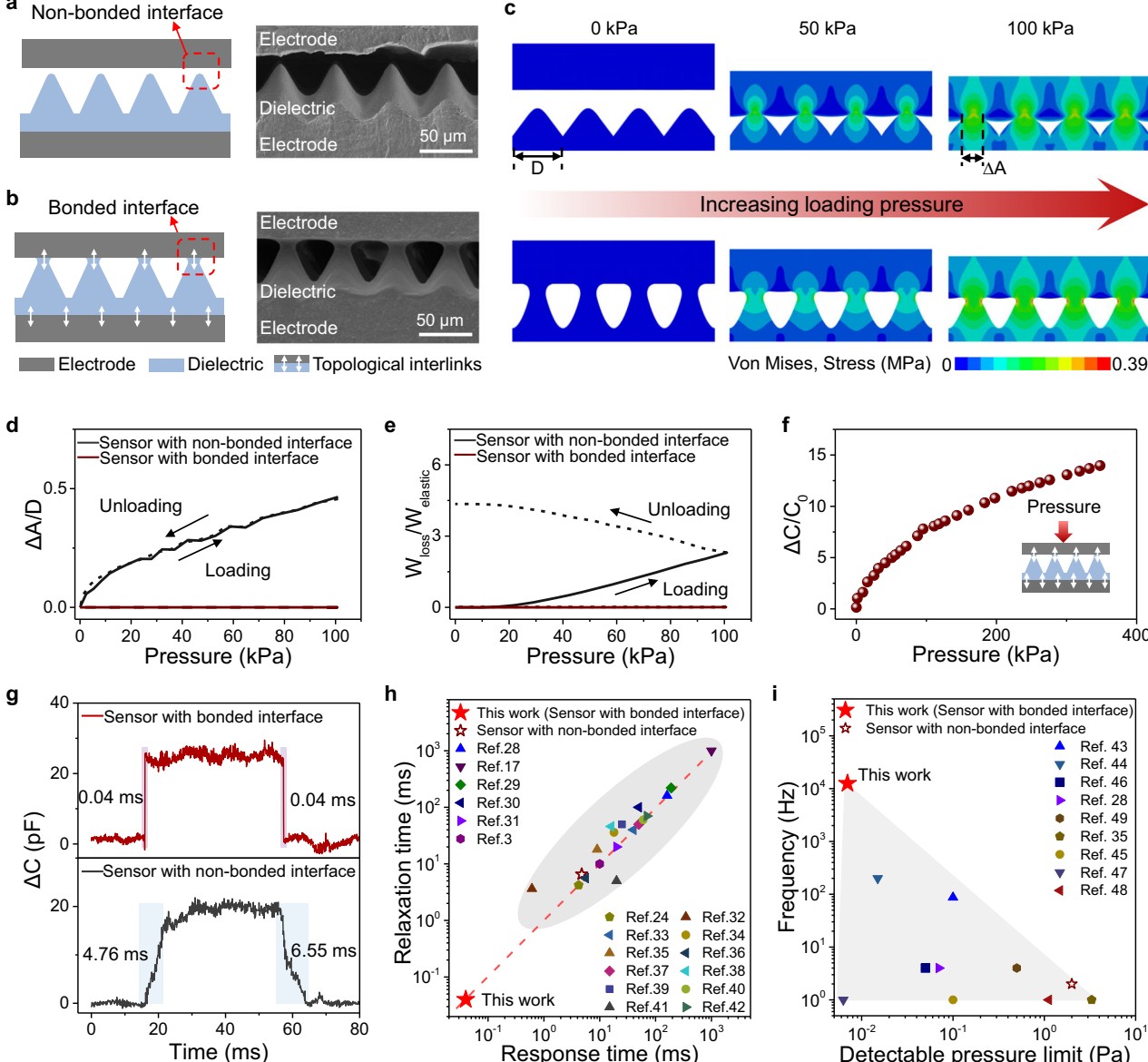

**Fig. 1 | Design of microstructured pressure sensors with a bonded interface via topological interlinks. a** Schematic and SEM image of the conventional microstructured pressure sensor with a non-bonded interface. **b** Schematic and SEM image of our pressure sensor with a bonded interface. **c** Finite element simulations of both non-bonded and bonded pressure sensors under the loading process. **d** Comparison of normalized increased contact area $\Delta A/D$ in a loading-unloading cycle. **e** Comparison of normalized energy loss in a loading-unloading cycle.

**f** Normalized change in capacitance as a function of pressure of the bonded pressure sensor. **g** The response-relaxation times of the sensor with a bonded interface and the sensor with a non-boned interface. **h** Comparison of our sensor and existing capacitive sensors in terms of response time and relaxation time. **i** Comparison of our sensor with existing capacitive sensors in terms of detectable pressure limit and corresponding frequency range.

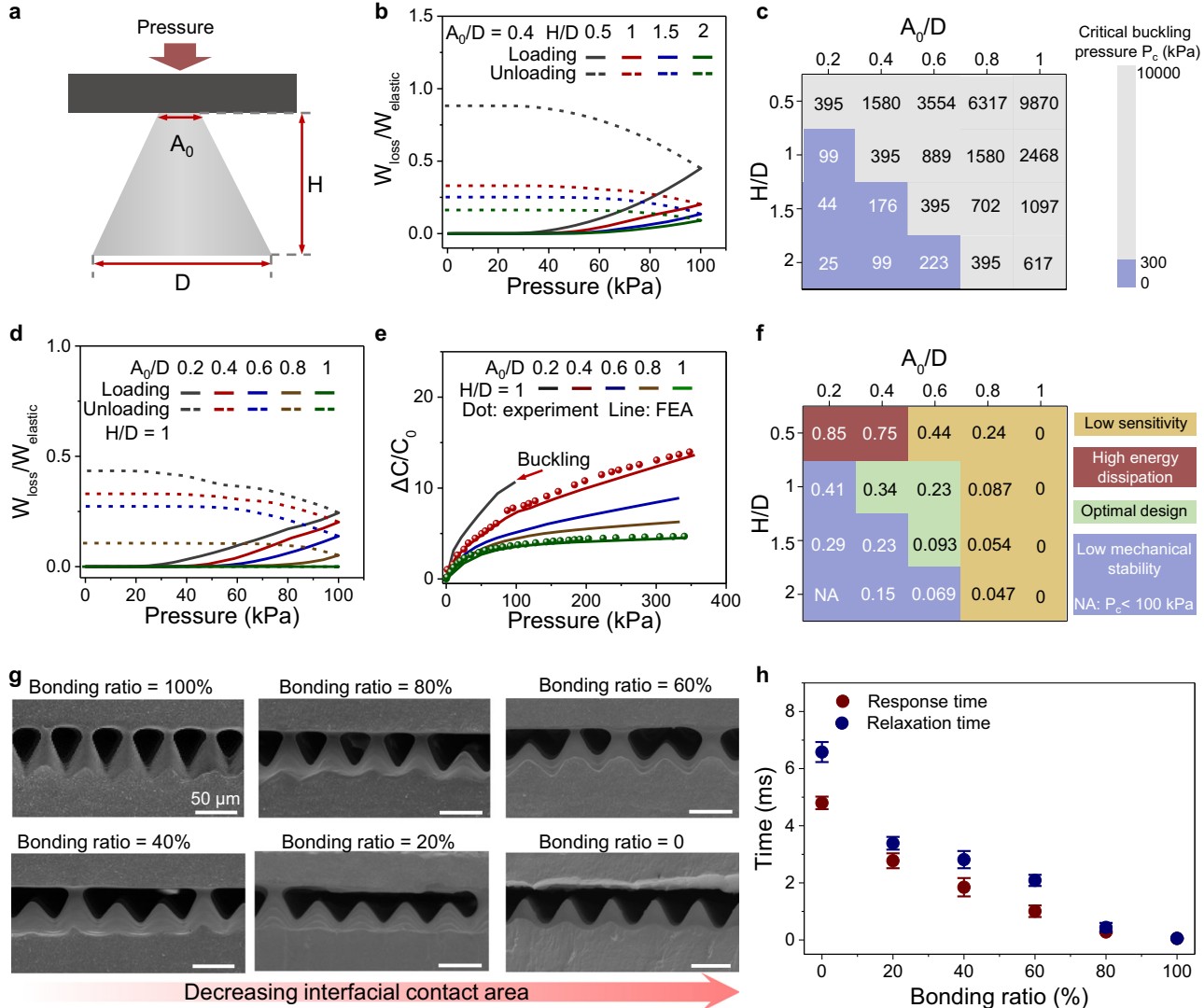

**Fig. 2 | Effect of microcone structure on response-relaxation time. a** Geometric parameters of the microcone. **b** Simulated energy loss versus pressure of microcones with various heights *H*. **c** Contour plot of critical buckling pressure of 20 microcones. **d** Simulated energy loss versus pressure of microcones with various bonded areas *A*. **e** Normalized capacitance change versus pressure of microcones

with various bonded areas. The dotted data are experimental results. **f** Contour plot of normalized energy loss of 20 microcones. **g** SEM of microcone arrays with different bonding ratios. **h** Response-relaxation time of microcones arrays with different bonding ratios. Error bars represent the standard deviation of three repeated measurements of the time.

lower $E''/E'$ as the doping wt.% increases. The high viscosity usually leads to a large adhesion force, thereby prolonging the separation (Supplementary Fig. 7). This phenomenon is further supported by the measured adhesion force of various materials using an atomic force microprobe (Fig. 3b). Additionally, the bonded microstructured interface contributes to the reduced interfacial adhesion, facilitating faster separation. We measured the adhesion strength between the 2 wt.% PDMS-CNTs dielectric and the electrode (7 wt.% PDMS-CNTs) under a 100 kPa load over 10 min. As shown in Fig. 3c, the adhesion strength of the bonded interface is negligibly small (-0.01 kPa), whereas the flat interface and the non-bonded interface exhibit much higher adhesion strength values of 28 kPa and 0.23 kPa, respectively.

The combination of low-viscosity 2 wt.% PDMS-CNTs dielectric with a bonded microstructured interface leads to ultrafast response and relaxation time of the sensor, as shown in Fig. 3d, e. Our sensor stands out substantially compared with sensors with higher viscosity and non-bonded interfaces. It achieves both response and relaxation times as low as 0.04 ms−this is an order of magnitude faster than its counterparts. Low viscosity and bonded interface also contribute to

low hysteresis and high mechanical stability of the sensor. Figure 3f, g shows substantially low hysteresis during a loading-unloading cycle to 100 kPa, and there is barely a capacitance-pressure hysteresis loop. The low hysteresis is related to the negligible viscous dissipation of the interface. Furthermore, the bonded microstructured interface exhibits a high interfacial toughness of 530 Jm$^{-2}$ measured by 180° peeling test (Supplementary Fig. 8). Such high interfacial toughness allows the sensor to work stably under certain harsh mechanical conditions. Figure 3h shows that the sensor exhibits a highly stable signal when subjected to repeated rubbing with a shear stress of ~45 kPa and a pressure of 200 kPa over a displacement of 2 mm for 10,000 cycles.

## Frequency bandwidth and frequency resolution of the pressure sensor

Sensors with rapid response and relaxation times can detect high-frequency vibrations. We use a testing system (Fig. 4a) to verify the frequency detection range. The experimental setup includes a computer-controlled function generator, a power amplifier, and a vibration generator. When vibrations of 12,500 Hz are generated, the

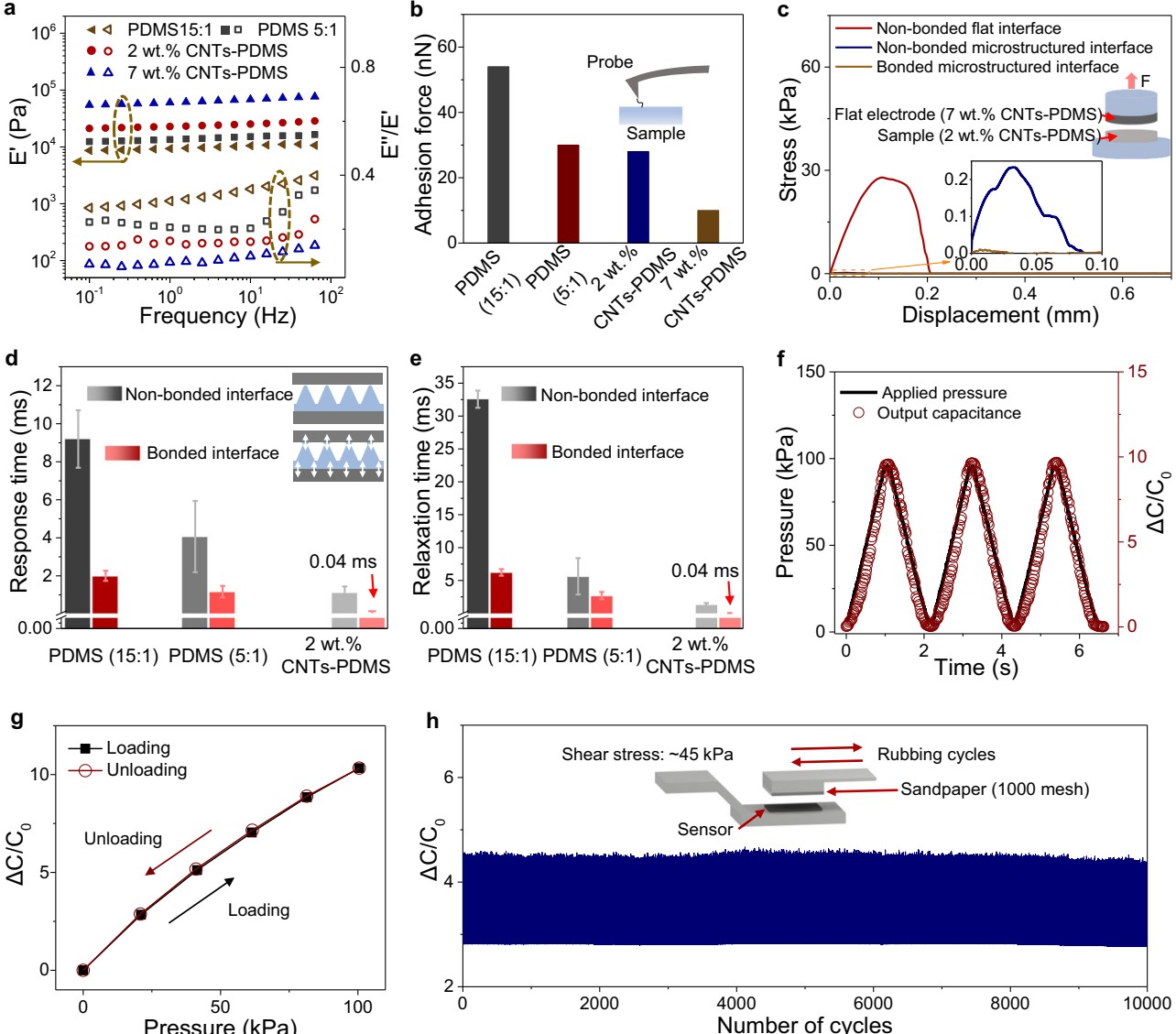

**Fig. 3 | Effect of material on response-relaxation time. a** Storage modulus ($E'$), loss modulus ($E''$) and their ratios of pure PDMS (with elastomer and curing agent ratio of 5:1), PDMS-CNTs composite (with 2 wt.% CNTs and 7 wt.% CNTs). **b** Adhesion force of pure PDMS (with elastomer and curing agent ratio of 15:1 and 5:1), PDMS-CNTs composite (with 2 wt.% CNTs and 7 wt.% CNTs). **c** The adhesion strength of 2 wt.% PDMS-CNTs with three different interfaces. **d** Response time of six sensors: sensors with non-boned interfaces consisting of three different dielectric layers made of pure PDMS (with elastomer to curing agent ratios of 15:1 and 5:1) and PDMS-CNTs composite and sensors with a bonded interface based on three different dielectric layers of pure PDMS (with elastomer to curing agent ratios of 15:1 and 5:1) and PDMS-CNTs composite. Error bars represent the standard deviation of three repeated measurements of the time. **e** Relaxation time of the six sensors mentioned in (**d**). **f** Dynamic response of the sensor as a function of time under a pressure of 100 kPa. **g** Normalized change in capacitance of the sensor during the loading and unloading cycle at a pressure of 100 kPa. **h** Response of the pressure sensor to cyclic rubbing (10,000 cycles) under a combination of pressure (200 kPa) and a shear (45 kPa). The friction test was performed by applying a sheet of #1000 sandpaper to the surface of the sensor.

sensor can detect the signal with a period of 0.08 ms (Fig. 4b, Supplementary Fig. 9), and the corresponding Fourier transform result shows a peak at 12,500 Hz (Fig. 4c). Furthermore, the sensor can detect signals of vibrations at different amplitudes at 12,500 Hz (Supplementary Fig. 10).

Sensors with a non-bonded interface have a much narrower frequency range compared to those with bonded interfaces. Figure 4d shows the capacitance response to vibrations with frequencies of 200, 400, 1000, 3000, 6000, 9000, and 12,000 Hz of the two sensors, which use the same materials and similar device configurations except that the former has bonded interfaces. Whereas the sensor with bonded interfaces can detect all those vibrations, the sensor with a non-bonded interface fails to detect vibrations of 400 Hz with high fidelity, as indicated in both time domain signals and corresponding frequency

domain analysis by short-time Fourier transform (Fig. 4e) or by fast Fourier transformation (Supplementary Fig. 11). Note that although the frequencies of 200 Hz and 400 Hz can be detected using the sensor with non-bonded interface, the output signals are distorted since $f_{max}$ of the sensor is only 73 Hz.

Our sensor also identifies high-frequency vibrations under pre-applied static pressure or low-frequency stimuli. At a static pressure of 100 kPa, we apply an extra vibrational signal of 500, 4000, 8000, and 12,500 Hz, and the sensor effectively detects the superimposed static and vibrational signals for all cases (Fig. 4f). Corresponding results in the frequency domain also well reflect vibrational signals of these frequencies (Supplementary Fig. 12). In addition, our sensor can detect vibrational stimuli with two substantially different frequencies (30 and 1000 Hz) and amplitudes (Supplementary Fig. 13), which may enable

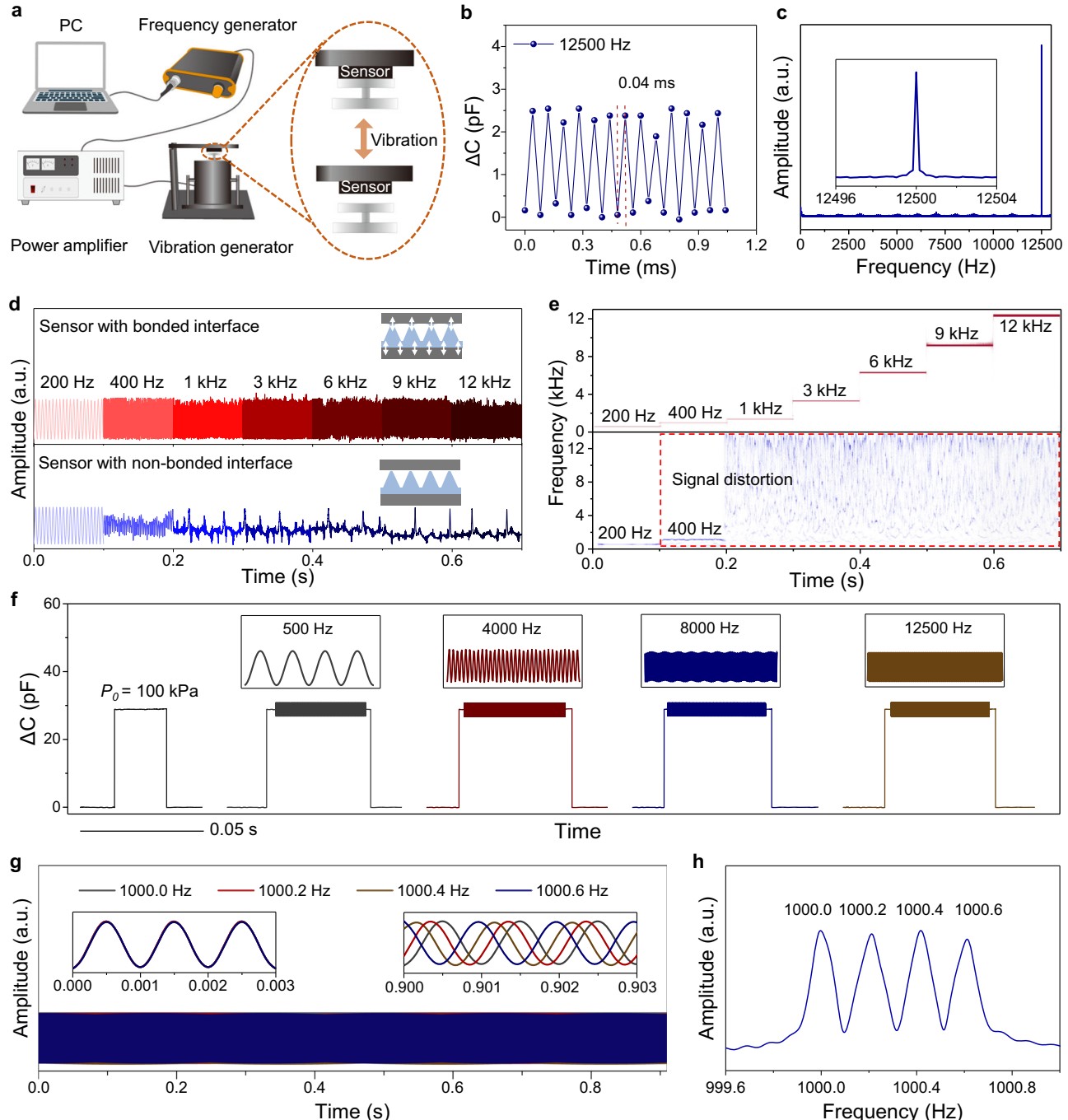

**Fig. 4 | Sensing performance of our sensor to high-frequency mechanical vibrations. a** Schematic of the wide-frequency mechanical stimuli detection and recording platform. **b** Capacitive response of the pressure sensor to the mechanical vibration at a fixed frequency of 12,500 Hz, showing the fast response time of our sensor. **c** The corresponding frequency spectra of the signals in (**b**). **d** Signal amplitude of the capacitive response to the vibration frequency from 200 Hz to 12,000 Hz between sensors with bonded (upper panel) and non-bonded (lower panel) interfaces. **e** Corresponding STFT spectrograms of (**d**). **f** Static pressure (-100 kPa) detection and vibrations detection with frequencies of 500, 4000, 8000, and 12,500 Hz under a static base pressure of -100 kPa. **g** Vibrational signals with frequencies of 1000.0, 1000.2, 1000.4, and 1000.6 Hz were detected using the pressure sensor. **h** Frequency spectra of the signals in (**g**).

the recognition of complex vibrational signals, such as sound recognition.

Frequency resolution is an index that determines the performance of the sensor to discriminate close frequencies. Figure 4g shows vibrational signals with frequencies of 1000.0, 1000.2, 1000.4, and 1000.6 Hz from the first to the 1000th period in the time domain detected using the sensor. Although the signals have close frequencies, they can be resolved in both the time domain and frequency domain (Fig. 4g), which clearly shows the peaks of 1000.0, 1000.2,

1000.4, and 1000.6 Hz. The result indicates a superior frequency resolution of 0.2 Hz at 1000 Hz, or $2 \times 10^{-4}$ (Fig. 4h).

## Sound detection using the sensor

We use the sensor in an artificial ear for sound detection. The human ear receives, transmits, and converts sound to physiological signals by the eardrum, the auditory ossicle, and the cochlea, respectively. Here, our artificial system uses a polyethylene naphthalate (PEN) membrane as the eardrum, the sensor as the cochlea, and a 3D printed aluminum

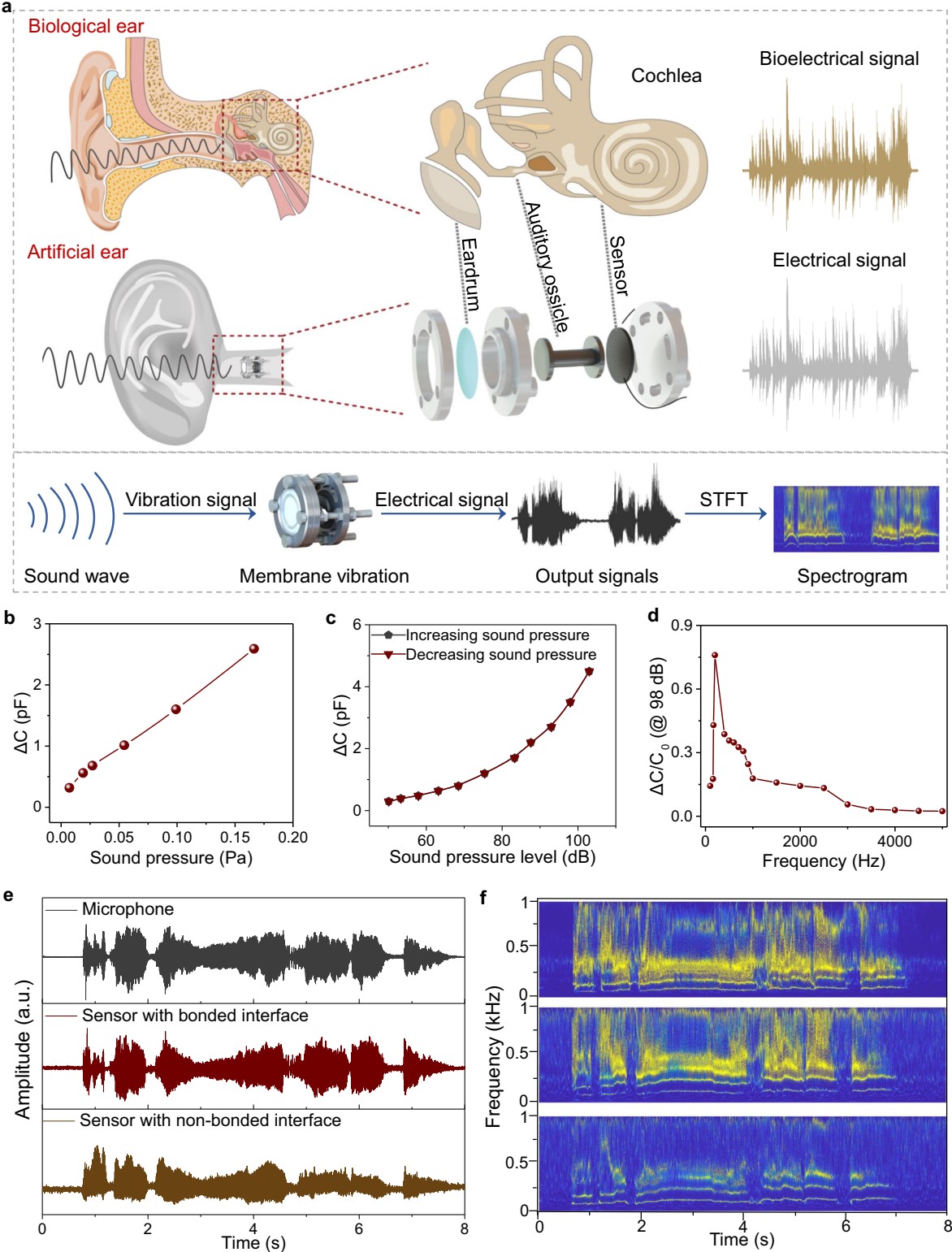

**Fig. 5 | Application of our sensor for sound detection. a** Schematics of the biological system and the artificial system for sound detection. **b** Capacitance change as a function of sound pressure. **c** Capacitance change as a function of sound pressure level. **d** Normalized change in capacitance of the system at different sound wave frequencies. SPL = 98 dB. **e** Acoustic waveforms recorded by a cellphone, by the sensor with a bonded interface and a non-bonded interface. **f** Corresponding STFT spectrograms of the acquired sound waveforms in (**e**).

auditory ossicle being connected to the artificial eardrum and the sensor, as shown in Fig. 5a. Sound waves cause the artificial tympanic membrane to vibrate, and the sensor converts the vibration to capacitance signals. The artificial tympanic membrane (area: 314 mm²) and

the aluminum auditory ossicle (with an end dimension of 4 mm) can also amplify the sound to enhance the pressure detection capability of the sensor. Figure 5b and Supplementary Fig. 14 show that the system can respond to a low sound pressure of 0.007 Pa, and the signal

amplitude ($\Delta C$, change in capacitance) increases linearly with increasing sound pressure in the range of 0.007–0.16 Pa. Similarly, at a given frequency (we select 200 Hz because it is the resonant frequency of the system, which will be discussed hereinafter), $\Delta C$ also increases monotonically with increasing sound intensity ranging from 50 to 110 dB because a higher sound intensity generates a larger sound pressure. Note that no hysteresis is observed during the process of increasing and decreasing sound intensity (Fig. 5c). Furthermore, at a fixed sound intensity of 98 dB, the system exhibits a nonlinear response with a frequency up to 5000 Hz, and a resonant frequency of 200 Hz is observed (Fig. 5d).

We have used our artificial ear for audio recording and compared the results with that of a commercially available microphone and that of the system using a non-bonded sensor. Figure 5e presents the recorded signals of the song "Bury Me Not On The Lone Prairie" using the microphone, the sensor with a bonded interface, and a sensor with a non-bonded interface. The waveforms of the first two are found to be generally consistent, while the waveform of the signal recorded using the sensor with a non-bonded interface is markedly distorted. Our wavelet transform analysis further verifies that the signals recorded using the microphone and the sensor with bonded interfaces are highly consistent in the frequency domain, whereas the signal recorded using the sensor with a non-bonded interface fails to record high-frequency information of a few hundred hertz as a result of the inability of the sensor to capture high-frequency vibrations (Fig. 5f). The results verify the potential of our sensor in acoustic applications. Although our artificial system can effectively showcase the practical feasibility of using this sensor in artificial ear applications, there is a large frequency-response variation due to the single structure. The frequency-response behavior can be optimized by integrating multiple tympanic membranes of different thicknesses or by incorporating frequency-dependent gain adjustments to compensate for the frequency-response variations.

## Discussion

In conventional capacitive soft pressure sensors, a common issue is the presence of gaps between the electrode and the viscoelastic dielectric layer. This often leads to high energy dissipation, resulting in response-relaxation times exceeding 10 ms. Such durations limit their effectiveness in detecting high-frequency signals. In our study, we address this limitation by reducing the viscosity of the PDMS dielectric through the incorporation of 2 wt.% CNTs. Additionally, we have engineered a bonding between microstructured microcones and the electrode, effectively reducing the response-relaxation time to ~0.04 ms.

We observed that the friction-induced energy dissipation is directly proportional to the contact area created during deformation. Our findings reveal that sharp microcones with larger bonded heads are more efficient in minimizing energy dissipation upon compression. However, a drawback is that these sharp microcones are prone to buckling instability, which can compromise mechanical stability. Furthermore, a larger bonded head can reduce the sensor's sensitivity. Through our analysis, we conclude that microcones with a moderate height and bonded head size offer an optimal balance, achieving rapid response-relaxation times, high mechanical stability, and enhanced sensitivity.

Our sensor design represents a significant advancement over existing PVDF-based flexible sensors, which may possess a broader frequency bandwidth[52]. Notably, our sensors can detect both static and dynamic pressures simultaneously. They are also softer and more flexible, broadening their practical applications in fields such as robotics, the metaverse, and biomedical engineering.

It is also important to highlight that standard commercial LCR meters may not be able to detect the limits of response time. For instance, the 6 ms response time of our previous sensors also represented the detection threshold of the LCR meter used[27]. Interestingly,

the remarkably short 0.04 ms response time of our current sensor is partly attributed to the 25,000 Hz bandwidth of our customized circuit board. This underscores the necessity of upgrading the testing systems in tandem with sensor improvements in the future.

Note that there are sensors that have similar or wider bandwidth ranges, including traditional silicon-based capacitive sensors[53], and flexible piezoelectric and triboelectric sensors[54,55]. However, the traditional silicon-based capacitive sensors are stiff, while the piezoelectric and triboelectric sensors are limited to the detection of dynamic signals. By contrast, our capacitive sensor is soft and has little signal drift[56]; it can not only detect static pressure but also record high-frequency vibrations over 10 kHz, making it potential for a wider range of applications.

## Methods

### Fabrication of the microstructured mold

A resin template with a microcone array with cone diameter of 50 μm and height of 40 μm was fabricated using high-precision 3D printing (NanoArch S130, BMF Precision Tech, Inc.). The density of the microcones is $4 \times 10^4$ cm$^{-2}$. The bonding joint is about 2 μm in diameter. A mixture of PDMS base and curing agent (Sylgard 184, Dow Corning Co., Ltd.) with a mass ratio of 5:1 was cast onto the microcone array mold. After curing at 80 °C for 30 min, the templated PDMS layer was peeled off, serving as a reverse template. Next, the reverse template was subject to air plasma treatment (TS-PL05, Dongxingaoke Co., Ltd) at 50 W for 3 min to realize an easy separation between the template and the PDMS-CNTs composite.

### Preparation of the PDMS-CNTs composites

The PDMS (Sylgard 184, Dow Corning Co., Ltd.) and CNTs (purity: 95%, diameter: 10–20 nm, length: 10–30 μm, Nanjing Xianfeng Nanotechnology Co., Ltd.) were dispersed in trichloromethane (Aladdin, 99%) for 2 h by ultrasonication. PDMS base and curing agent with a base-to-curing agent ratio of 5:1 and CNTs (2 wt.% or 7 wt.%) were directly mixed to form a homogeneous PDMS-CNTs solution. Next, the solution was cast on a specific mold. After the trichloromethane was evaporated, the degassed PDMS-CNTs composites were cured at 80 °C for 2 h. For the electrode, the weight ratio of CNTs to PDMS was controlled to be 7 wt.% with 50 μm thickness. For the dielectric, the uncured PDMS-CNTs solution was cast on the reverse template. The weight ratio of CNTs to PDMS was controlled to 2 wt.% with a 100 μm thickness.

### Preparation of the pressure sensor

PDMS base (10 g) and curing agent (2 g) were added to (100 mL) trichloromethane, and the mixture was sonicated for 30 min to form a solution. The pre-prepared electrodes and dielectric layer were then immersed in the solution for 6 h at room temperature for swelling. Next, the swollen top electrode, microstructured dielectric, and bottom electrode were stacked in sequence and pressed under 20 kPa to allow for curing to form bonded interfaces. The trilayer was cut to be 10 mm × 10 mm in area and used as a sensor.

### Characterization and measurements

The morphologies of the sensor were characterized using a field-emission scanning electron microscope (TESCAN MIRA3). The external pressure was applied and measured using a force gauge controlled with a computer-programed stage (XLD-500E, Jingkong Mechanical Testing Co., Ltd). The pull-off force test was tested by atomic force microscopy (Nanoscope IIIa, Multimode, Picoforce force control module, DI-Veeco Instruments Inc.) in the tapping mode. A rheometer (TA Q850) was used to carry out dynamic mechanical analysis in a tension mode. The storage modulus $E'$ and the loss modulus $E''$ were measured as a function of temperature. The capacitance response was measured using an LCR meter (E4980AL, KEYSIGHT) at a testing

frequency of 1 MHz if not specified. The loading and unloading cycles were conducted at a loading rate of 100 kPas$^{-1}$. Unless a preload is specified, a vibration generator (Model BL-ZDQ-2185, Hangzhou Peilin Instrument Co. Ltd.) was adopted to apply vibration of given frequencies to the sensor. The capacitance responses for response time, relaxation time, and mechanical vibration were measured using a home-made circuit with a sampling frequency of 25 kHz. Adhered samples of 10 mm in width and 30 mm in length were used to measure the interfacial toughness of the sample, tested through a standard 180° peel test with a force gauge and computer-controlled stage (XLD-500E, Jingkong Mechanical Testing Co., Ltd). The peeling speed was maintained at 50 mmmin$^{-1}$. The sensors were affixed to a polyimide sheet using a cyanoacrylate glue (Krazy Glue) serving as a rigid backing.

### Detection of sound

The artificial ear was designed following the structure of the human ear, consisting of a tympanic membrane, an auditory ossicle, and our sensor. The auditory ossicle was fabricated by 3D printing of aluminum alloy. The tympanic membrane was made of a fluorinated ethylene propylene film with low damping (diameter: 50 mm). An aluminum-alloy rod of dumbbell shape was designed to mimic the auditory ossicle. The acoustic measurement was conducted in an acoustic chamber to ensure immunity from environmental noise interference.

For the acoustic sound detection, we used a computer-controlled speaker as an acoustic source. The sound pressure level produced by the speaker was measured using a sound meter. The sound pressure $P$ is correlated to $SPL$ and expressed as $SPL = 20\lg (P/P_0)$, where $P_0 = 20\,\mu Pa$ is the reference sound pressure in the air. The response as a function of sound pressure and the capacitance changes with increasing sound intensity ranging from 50 to 110 dB were measured at 200 Hz.

### Reporting summary

Further information on research design is available in the Nature Portfolio Reporting Summary linked to this article.

## Data availability

The data generated in this study are provided in the Main Text and the Supplementary Information. Additional data are available from the corresponding author upon request.

## Code availability

The code that supports the fundings of this study is available in GitHub with the identifier https://github.com/tao-sun2/SUSTECh-frequency-domain-diagrams.

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

## Acknowledgements
The work was funded by the National Natural Science Foundation of China (No. T2225017 and 52073138), the Science Technology and Innovation Committee of Shenzhen Municipality (No. JCYJ20210324120202007), the Shenzhen Sci-Tech Fund (No. YTDPT20181011104007), and the Guangdong Provincial Key Laboratory Program (No. 2021B1212040001).

## Author contributions
C.F.G. and Y.Z. conceived the idea, C.F.G., Y.C., and L.W. directed the study. Y.Z. conducted most of the experiments. X.Z. designed and manufactured the circuit board. L.W. and N.Z. performed the finite element analysis. J.Z. and N.B. wrote the code of MATLAB for acquiring frequency-domain diagrams. T.S. performed the recognition of acoustic sound detection. X.H., G.L., and L.Z. also participated in the experiments. C.F.G., L.W., and Y.Z. wrote the manuscript. All authors contributed to the writing of the final version of the manuscript.

## Competing interests
The authors declare no competing interests.
