## [Peer Review File · Nature Communications]

REVIEWER COMMENTS

Reviewer #1 (Remarks to the Author):

In the manuscript, the authors report on a capacitive sensor with a 0.1 ms response/relaxation time, or 5000 Hz frequency bandwidth that enables high-frequency vibrations detection, including acoustic applications. This is a big breakthrough in flexible electronics because capacitive pressure sensors are a landmark element of the field but fail to respond to high-frequency dynamic stimuli. The authors offer deep understanding on the response-relaxation speed of sensors. They point out that the slow response speed in conventional capacitive sensors or e-skins comes from the interfacial behavior, a factor that has not been noticed before. The interfacial contact and separation process dissipate energy and take time to respond to stimuli or to recover, and they use a bonded interface to eliminate the contact-separation and thus reduce the viscous energy dissipation, and eventually, improve the response-relaxation time from ~10 ms to 0.1 ms. Overall, this work is novel and important, and the results are interesting. I therefore strongly recommend for publication of this work in Nature Communications. There are also a few points that may help further improve the work, or the authors can consider in their follow-up studies.

1. The low hysteresis is very interesting and important to the rapid response. I believe that the low hysteresis is also related to the reduced interfacial energy dissipation. This might also be a topic that worth a piece of separate work in the future. Now my question is: what is the loading rate for this data? Hysteresis is greatly affected by loading rate and the information should be given.
2. Please change “mechanical propertied” to “mechanical properties”.
3. I think PDMS/CNT should be changed to PDMS-CNT. The sign “/” means “or”.
4. Please change “elongation” in Figure 1b to “elongate”.
5. Can the authors discuss further strategies (or barriers) to improve the response-relaxation speed? It seems that the theoretical limit is on nanoseconds because soft materials can respond that fast.
6. I suggest that the authors change “0.1 s response time” to “0.1 ms response/relaxation time” in the title. If relaxation is slow then the sensor can still not respond to 5000 Hz vibrations.

Reviewer #2 (Remarks to the Author):

The authors have presented an innovative structurally engineered pressure sensor with a unique architecture that showcases a short response-relaxation time, making it capable of detecting high-frequency signals. While the analysis of response and relaxation mechanisms is exhaustive, there are several concerns regarding the sensor's design, characterization, and experiments. The reviewer recommends revising the manuscript, addressing the concerns listed below to further improve the manuscript quality.

Major:

1. The authors claimed that contact-separation behavior between the microcone tips and the flat elastomer layer is a significant cause for the long response-relaxation time. However, the microcone structure's impact is not well characterized.

(1) If the contact area of the cone tips is increased, will the response-relaxation time be further reduced

in the bonded case? In the unbonded case, supposedly increasing the interfacial contact area would facilitate the strain energy dissipation. Simulation or experiments to explore the relationships between contact area and response time (and even dissipated energy) may provide clearer physical pictures.

(2) If the answer to the above question regarding the bonded case is positive, why not replace the cone with cylindrical dielectric pillars? The pillars have larger contact areas, which could prevent the contact-separation behavior from happening. If the answer to the question is negative, the authors are suggested to elaborate on how the cone-shaped structure accelerates the relaxation of the structure than the cylindrical case. A strain analysis during the response-relaxation process (at the very least in simulation) would be appreciated.

(3) Will the sharpness of the cone impact the response-relaxation time? Simulation or experiments that can reveal the cone sharpness and relaxation time relationship would better justify the design parameter of the microcone array.

(4) Changing the tip contact area could also influence the detectable pressure range. Modeling this effect could help justify the microcone design.

2. The authors claimed that the 0.1 ms response and relaxation time in Fig. 1 is the limitation of the test system, but not of the sensor. However, the test system shown in Supplementary Fig. 2 and 3 does not indicate any limitations.

(1) Is the ADC sampling rate of the MCU the limiting factor? The reviewer would suggest the authors clarify the true limiting factor rather than simply provide schematics of the circuits.

(2) If the circuit system is the bottleneck, the authors are suggested to use a better system to further investigate the true speed of the pressure sensor.

3. The sampling rate of the system appears to be varying during the author's experiments. In Fig. 1f, Fig. 2f, and Fig. 3b,g, the sampling frequencies appear to be different, causing different levels of details revealed in those figures. Please state clearly the experiment setup in each of the cases, and other cases mentioned in the paper if not listed here.

4. The author claimed that the reported pressure sensor could detect high-frequency responses. However, there are other flexible materials or flexible sensor structures that can detect MHz-level vibrations, such as PVDF and micromachined ultrasound transducers (DOI: 10.1038/s41378-018-0022-5). The advantage of the reported pressure sensor over these ultrasonic transducers remains unclear.

5. The artificial ear sensor has a resonance frequency of 200 Hz, and the response to high-frequency vibrations (>1000 Hz) is ~4 times weaker. This may compromise the sound recording fidelity. Can the pressure sensor, eardrum or ear canal be further engineered to optimize the sensor response to the high-frequency band?

6. The authors used carbon nanotubes as the doping material. The reviewer would appreciate a brief justification of the preference for CNTs over other nanomaterials that may obtain similar functions.

Minor:

7. On Page 6, Line 112, the unit 'ms' should be removed from the equation to ensure the unit of f_{max} is Hz.

Reviewer #3 (Remarks to the Author):

This manuscript presents an ultrafast piezocapacitive pressure sensor with a bonded microstructure.

Compared to conventional piezocapacitive type pressure sensor, this one exhibits a rapid response time of 0.1 ms, in contrast to the conventional tens of ms. This performance was achieved using a bonded, microstructured interface, which reduces interfacial elastic instability and viscoelastic dissipation. Due to its high response and relaxation speeds, it can detect not only dynamic forces but is also suitable for acoustic applications. However, the novelty and innovation of this article are not sufficient for publication in Nature Communications. A similar sensor structure and working mechanism have already been reported in Nature Communications by the same authors. There was only a minor difference compared to the previous publication (Nat. Commun. 2022, 13, 1317) Therefore, I do not recommend the publication of this manuscript in Nature Communications.

1. In the author's previous paper (Nat Commun. 2022, 13, 1317), the device closely resembles the one presented in this work. Even though both devices share the same material composition, structure, mechanism, and the idea of topological interlinks, they yield different results. The points of difference are as follows.

- Both devices feature the same topological interlinks and structure, yet they differ in interfacial toughness.
- In this manuscript, the authors achieve reduced response and relaxation times due to the topological interlinks. However, in the prior paper, despite the presence of these interlinks, the device exhibited a significantly longer response and relaxation time (~ 6 ms).
- The device in this manuscript has a reduced sensitivity to the pressure compared to the one in the previous study.
- The device in this manuscript has a lower detection limit for pressure than the one in the previous study.

2. In Fig. 4d, the device displays a resonant frequency at 200Hz and exhibits a nonlinear response up to 5000Hz. However, in Fig. 3d, 3f, 3g, the signal maintains the same amplitude across different frequencies. Why doesn't the signal exhibit frequency dependence?

3. In Fig. 4a, there's a depiction of the sensor output signal processing using deep learning for phrase reconstruction. However, within the main content of this paper, only some STFT data is provided for signal processing, and there is no mention or elaboration on signal processing and reconstruction through deep learning as indicated in the figure. Consequently, the section of the figure pertaining to signal processing and reconstruction should be removed.

4. In the study, the author employed 2 wt% and 7 wt% CNT/PDMS composites for the dielectric layer and electrode layer, respectively. To bolster the integrity and rationale of this work, it would be beneficial if the author provided justifications for selecting these specific concentrations. For instance, data or references on electrical conductance, dielectric constant, and dielectric loss could substantiate the author's choice of material composition.

5. It is recommended to add data regarding the characteristics of the microstructure, specifically related to its design, density, or the extent of the interlinked area

6. On page 9, the text states, "Our sensor also identifies high-frequency vibration under a pre-applied static pressure or low-frequency stimuli." How does the sensor perform when there is no pre-applied static pressure or low-frequency stimuli?

Response to reviewers for manuscript NCOMMS-23-38129A

We thank all reviewers for your constructive comments and suggestions to improve our work. In this letter, your comments are addressed carefully in a point-by-point manner. For your convenience, the modifications in the revised manuscript are marked in blue. To better help you understand the improvement of our work, we would like to give major modifications before point-by-point responses.

Major modification #1

We now clearly describe the primary reasons that limit the response-relaxation speed of conventional flexible piezocapacitive pressure sensors and the strategy to overcome such a limitation. In the revised **Introduction**, we write:

“While elastic elastomers can respond to mechanical stimuli in nanoseconds,¹⁷⁻¹⁹ conventional piezocapacitive flexible pressure sensors often exhibit a response-relaxation time on the level of tens of milliseconds, corresponding to a narrow frequency range up to tens of hertz. This low response-relaxation speed is primarily attributed to energy dissipation associated with viscoelastic materials and interfacial frictions. Soft dielectrics are typically viscoelastic materials that dissipate energy during loading-unloading cycles. Such an energy loss gets more pronounced when softer materials are used for detecting subtle pressures^{20,21} Also, during the contact-separation process, the interfacial friction and adhesion between the electrode and dielectric further contribute to energy loss.^{22,23} To improve the response-relaxation speed, a common strategy is to engineer the dielectric layer with microstructured surfaces since Bao et al. in 2010.²⁰ This strategy works through two principles. First, the microstructures reduce the bulk viscoelasticity of the dielectric by storing more elastic energy in smaller deformations. Second, they reduce the contact area between the dielectric and electrode, thereby lowering energy dissipation due to interfacial friction and adhesion. However, despite the reduced energy dissipation by introducing microstructures, the response-relaxation time remains largely above 1 ms to date. This limitation seems to be unreconcilable as long as viscoelastic materials are used and interfacial gaps persist.”

“In this work, we present a strategy for downscaling the response or relaxation time of flexible piezocapacitive pressure sensors to ~0.04 ms by seamlessly bonding a low-viscosity microstructured dielectric with the electrode. The dielectric is made by dispersing 2 wt.% carbon nanotubes (CNTs) within a polydimethylsiloxane (PDMS) matrix which reduces the material viscosity and surface adhesion. Without interfacial gaps, the bonded microstructured interfaces substantially diminish the friction-induced energy dissipation. We show that our sensor can quickly respond to stimuli from steady pressure to high-frequency vibrations over 10 kHz.”

Major modification #2

We performed finite element simulations to elucidate the energy dissipation of the bonded and non-bonded pressure sensors. Revised Fig. 1c-e clearly suggest that the energy loss during a loading-unloading process of the bonded pressure sensor is significantly smaller than that of the non-bonded counterpart.

Fig. 1. | Design of microstructured pressure sensors with bonded interface via topological interlinks. **a**, Schematic and SEM image of the conventional microstructured pressure sensor with a non-bonded interface. **b**, Schematic and SEM image of our pressure sensor with bonded interface. **c**, Finite element simulations of both non-bonded and bonded pressure sensors under a loading-unloading cycle. **d**, Comparison of normalized increased contact area $\Delta A/D$ in a loading-unloading cycle. **e**, Comparison of normalized energy loss in a loading-unloading cycle.

On Page 5, we add:

“Conventional microstructured pressure sensors have a gap between the electrode and the dielectric (Fig. 1a), while our sensors have a bonded electrode-dielectric interface (Fig. 1b). To compare their different response-relaxation behavior, we fabricated two sensors. The top and bottom electrodes are made by dispersing 7 wt.% carbon nanotubes (CNTs) in polydimethylsiloxane (PDMS), while the middle dielectric layer is made of PDMS with 2 wt.% CNTs fillers. Microstructured dielectric layers were fabricated by molding a 3D-printed template of microcones (Supplementary Fig. 1). The bonded interface is further realized by immersing electrodes and dielectric in trichloromethane solvent with uncured PDMS networks, followed by a curing process to form topological interlinks (Supplementary Fig. 2).²⁷

To elucidate the reduced energy dissipation of bonded interfaces, we performed finite element analysis (FEA) in which two sensors are compressed to 100 kPa and then recovered (Fig. 1c). The bottom area of the microcone and the increased electrode-dielectric contact area is denoted as D and ΔA , respectively. For the non-bonded sensor with a gap, ΔA increases quickly upon loading, as shown in Fig. 1d, and the stress concentrates at contacted regions (Fig. 1c). As a result, the friction contributes to substantial energy loss (denoted as W_{loss}) that is higher than the maximum elastic energy (denoted as W_{elastic}) during loading-unloading process (Fig. 1e). On the contrary, the bonded pressure sensor only shortens under compression with negligibly smaller ΔA than that of the non-bonded counterpart

(Fig. 1d). Therefore, the energy loss is significantly reduced (Fig. 1e), leading to an elevated response-relaxation time of the bonded sensor.”

Major modification #3

We added both experiments and finite element simulations to investigate the effect of different structures on the response-relaxation time. Results are now presented in a newly added figure (Fig. 2).

Fig. 2 Effect of microcone structure on response-relaxation time. **a**, Parameters of microcone’s structure. **b**, Simulated energy loss versus pressure of microcones with various heights H . **c**, Contour plot of critical buckling pressure of 20 microcones. **d**, Simulated energy loss versus pressure of microcones with various bonded areas A . **e**, Normalized capacitance change versus pressure of microcones with various bonded areas. The dotted data are experimental results. **f**, Contour plot of normalized energy loss of 20 microcones. **g**, SEM of microcone arrays with different bonding ratios. **h**, Response-relaxation time of microcones arrays with different bonding ratios.

On Page 6, we add:

Effect of microcone structure on response-relaxation time

The bonded microcones play a key role in reducing the energy dissipation during the contact-separation

process, thus increasing the response-relaxation speed. The effect of microcone structure on the response-relaxation time is further investigated by FEA. The structure of a single microcone in 2D configuration can be described by three parameters: height H , top surface A_0 (i.e., initial contact area), and bottom surface D (Fig. 2a). 20 different microcones are investigated by varying $A_0/D=0.2, 0.4, 0.6, 0.8, 1$ and $H/D = 0.5, 1, 1.5, 2$. With a fixed $A_0/D = 0.4$, we first study the effect of height H on the energy dissipation by varying H/D . Results in Fig. 2b show that sharper microcones (i.e., larger H/D) show less energy loss. However, sharp microcones may also undergo buckling instability that undermines the mechanical stability of the sensor. For a tapered column with clamped top/bottom surfaces, the critical buckling force can be calculated by $\frac{4\pi^2 EI_D}{H^2} \frac{A_0^2}{D^2}$ where E is the Young's modulus and $I_D = \frac{\pi D^4}{64}$.⁵⁰ The critical buckling pressure (denoted as P_c) of 20 microcones are summarized in Fig. 2c. Considering that the maximum applied pressure should be greater than 300 kPa, microcones with small A_0/D and large H/D should not be adopted (marked with blue in Fig. 2c). Second, we analyze how initial contact area A_0 affect the energy dissipation by setting $H/D = 1$ and varying A_0/D . The normalized energy loss in Fig. 2d suggests that larger A_0/D yields a lower energy loss. This conclusion is expected because stout microcones (i.e., larger A_0/D) create less contact area under the same pressure while preserving more elastic energy W_{elastic} . However, we may not rush to conclude that microcone with $A_0/D = 1$ (i.e., a cylindrical micropillar) is the best by far, because the flat dielectric is notorious for its low sensitivity,²⁰ i.e., the slope of $\Delta C/C_0$ versus pressure curve. As also revealed by our models in Fig. 2e, we show that the sensitivity gradually drops when A_0/D increases (Supplementary Text 2), leading to an undermined detection range due to the saturated signals, as validated by experiments (dotted data in Fig. 2e). Therefore, the tradeoff between response-relaxation speed and sensitivity should also be considered. By summarizing the normalized energy loss of 20 microcones in Fig. 2f with background colors indicating other metrics, we show that microcones with moderate values of $(A_0/D, H/D)=(0.4, 1), (0.6, 1)$ and $(0.6, 1.5)$ can simultaneously achieve low energy dissipation, high sensitivity, and high mechanical stability. Note that it is difficult to fabricate bonded microcones with perfect geometries, and microcones used in this work have close values of $(A_0/D, H/D) \approx (0.4, 1)$ (see SEM in Fig. 1b). In addition to the structure of microcones, the bonding ratio of microcone arrays also affect the response-relaxation time of the sensor. In this regard, we compare the performance of six sensors with bonding ratios of 0, 20, 40, 60, 80, and 100% (Fig. 2g). Results in Fig. 2h manifest that the 100% bonded ratio provides the lowest response-relaxation time of ~ 0.04 ms, which is higher than previous sensors without fully bonded interfaces (Supplementary Fig. 5).²⁷

Major modification #4

We have updated the circuit board of the customized testing system, increasing the sampling frequency bandwidth from 10 kHz to 25 kHz. The detailed design of the circuit board is now presented in **Supplementary Fig. 3** and measured data are correspondingly modified in main text.

(1) The resistance R_4 connected to pin 3 (OUT) of the precision timer (SE555) was adjusted from 51 Ω to 10 Ω to accommodate the measured pulse waveform under high frequency, thereby shortening the rise time of the high level.

(2) The capacitance C_2 connected to pin 7 (Cap+) of the SE555 timer was reduced from 100 pF to 10 pF to align with software modifications, controlling the charging time of the tested capacitor within 25

μs .

(3) The range-switching module of the original circuit was removed, and a fixed range connected to the tested capacitor with the shortest PCB trace path was added. The primary objective is to reduce the lead length of the capacitor measurement circuit, minimize external interference noise picked up along the high-frequency analog signal traces, and enhance the accuracy and stability of capacitance measurements.

(4) The triggering interval of the SE555 timer was reduced from 100 to 40 μs , and the trigger pulse width was decreased from 20 to 12 μs . Simultaneously, the charging time of the tested capacitor was further shortened to within 25 μs , allowing an increase in the sampling frequency bandwidth from 10 to 25 kHz. Additionally, precise equipment such as a signal generator and a high-speed oscilloscope were employed to accurately measure the delay t_d between the trigger pulse and the beginning of charging. Compensation and calibration for this delay were achieved through software algorithms to mitigate additional measurement errors arising from the increased sampling frequency.

Supplementary Fig. 3. Schematic design diagram of the digital circuit board. a, Schematic diagram of capacitor charging and discharging circuit and the correspondence between capacitance value and charging time. b, The capacitance measurement circuit that is mainly composed of five modules: power supply, range switching, capacitance measurement, microcontroller, and high-speed communication. c, Schematic diagram of capacitor charging and discharging circuit and the correspondence between capacitance value and charging time.

Point-by-point responses to Reviewers

Reviewer #1

General comments:

In the manuscript, the authors report on a capacitive sensor with a 0.1 ms response/relaxation time, or 5000 Hz frequency bandwidth that enables high-frequency vibrations detection, including acoustic applications. This is a big breakthrough in flexible electronics because capacitive pressure sensors are a landmark element of the field but fail to respond to high-frequency dynamic stimuli. The authors offer deep understanding on the response-relaxation speed of sensors. They point out that the slow response speed in conventional capacitive sensors or e-skins comes from the interfacial behavior, a factor that has not been noticed before. The interfacial contact and separation process dissipate energy and take time to respond to stimuli or to recover, and they use a bonded interface to eliminate the contact-separation and thus reduce the viscous energy dissipation, and eventually, improve the response-relaxation time from ~10 ms to 0.1 ms. Overall, this work is novel and important, and the results are interesting. I therefore strongly recommend for publication of this work in Nature Communications. There are also a few points that may help further improve the work, or the authors can consider in their follow-up studies.

Response: We thank the reviewer for your positive comments on our work.

Comment #1

The low hysteresis is very interesting and important to the rapid response. I believe that the low hysteresis is also related to the reduced interfacial energy dissipation. This might also be a topic that worth a piece of separate work in the future. Now my question is: what is the loading rate for this data? Hysteresis is greatly affected by loading rate and the information should be given.

Response #1

We agree with the reviewer that low hysteresis is also related to reduced energy dissipation. This is indeed a good topic for the future study. The answer to your question: the loading rate is $10 \text{ N}\cdot\text{s}^{-1}$ and the information has been added in the “**Methods**” section in the revised manuscript.

On Page 15, we add the following description in **Methods**:

“The loading and unloading cycles were conducted at a loading rate of $100 \text{ kPa}\cdot\text{s}^{-1}$.”

Comment #2

Please change “mechanical propertied” to “mechanical properties”.

Response #2

We thank the reviewer for pointing our typos. They are now corrected.

Comment #3

I think PDMS/CNT should be changed to PDMS-CNT. The sign “/” means “or”.

Response #3

We have adopted “**PDMS-CNTs**” in the revised manuscript per the suggestion of the reviewer.

Comment #4

Please change “elongation” in Figure 1b to “elongate”.

Response #4

Figure 1 has been revised. Typos are carefully checked.

Comment #5

Can the authors discuss further strategies (or barriers) to improve the response-relaxation speed? It seems that the theoretical limit is on nanoseconds because soft materials can respond that fast.

Response #5

Thank you for your suggestion. Discussions are added on Page 13:

“We observed that the friction-induced energy dissipation is directly proportional to the contact area created during deformation. Our findings reveal that sharp microcones with larger bonded heads are more efficient in minimizing energy dissipation upon compression. However, a drawback is that these sharp microcones are prone to buckling instability, which can compromise mechanical stability. Furthermore, a larger bonded head can reduce the sensor’s sensitivity. Through our analysis, we conclude that microcones with a moderate height and bonded head size offer an optimal balance, achieving rapid response-relaxation times, high mechanical stability, and enhanced sensitivity.”

“It’s also important to highlight that standard commercial LCR meters may not be adequate for detecting the limits of response time. For instance, the 6 ms response time of our previous sensors also represented the detection threshold of the LCR meter used.²⁷ Interestingly, the remarkably short 0.04 ms response time of our current sensor is partly attributed to the 25 kHz bandwidth of our customized circuit board. This underscores the necessity of upgrading the testing systems in tandem with sensor improvements in the future.”

Comment #6

I suggest that the authors change “0.1 s response time” to “0.1 ms response/relaxation time” in the title. If relaxation is slow then the sensor can still not respond to 5000 Hz vibrations.

Response #6

Thanks for the suggestion. The title is changed to “**Ultrafast piezocapacitive soft pressure sensors with over 10 kHz bandwidth via bonded microstructured interfaces**”.

Reviewer #2

General comments.

The authors have presented an innovative structurally engineered pressure sensor with a unique architecture that showcases a short response-relaxation time, making it capable of detecting high-frequency signals. While the analysis of response and relaxation mechanisms is exhaustive, there are several concerns regarding the sensor's design, characterization, and experiments. The reviewer recommends revising the manuscript, addressing the concerns listed below to further improve the manuscript quality.

Response: We sincerely thank the reviewer for recognizing our work as “innovative”. We have added both theoretical modeling, finite element analysis, and experiments to address your comments.

Comment #1.

The authors claimed that contact-separation behavior between the microcone tips and the flat elastomer layer is a significant cause for the long response-relaxation time. However, the microcone structure's impact is not well characterized.

(1) If the contact area of the cone tips is increased, will the response-relaxation time be further reduced in the bonded case? In the unbonded case, supposedly increasing the interfacial contact area would facilitate the strain energy dissipation. Simulation or experiments to explore the relationships between contact area and response time (and even dissipated energy) may provide clearer physical pictures.

(2) If the answer to the above question regarding the bonded case is positive, why not replace the cone with cylindrical dielectric pillars? The pillars have larger contact areas, which could prevent the contact-separation behavior from happening. If the answer to the question is negative, the authors are suggested to elaborate on how the cone-shaped structure accelerates the relaxation of the structure than the cylindrical case. A strain analysis during the response-relaxation process (at the very least in simulation) would be appreciated.

(3) Will the sharpness of the cone impact the response-relaxation time? Simulation or experiments that can reveal the cone sharpness and relaxation time relationship would better justify the design parameter of the microcone array.

(4) Changing the tip contact area could also influence the detectable pressure range. Modeling this effect could help justify the microcone design.

Response #1

Thank you for your insightful comments. To elucidate the structure effects on the performance of the sensor, we have carried out both theoretical modeling and finite element analyses of 20 different microcones. Results are now presented in the newly added **Fig. 2** and the main text “**Effect of microcone structure on response-relaxation time**”. Brief answers to your questions are as follows:

(1) For a bonded interface, increasing initial contact area A_0 will reduce energy dissipation during the contact-separation process, thus increasing response-relaxation time.

- (2) However, a large initial contact area also reduces the sensitivity of the sensor. It is widely known that a cylindrical micropillar (like a flat dielectric) has a low sensitivity. Instead, microcones and pyramids are adopted to elevate sensitivity.
- (3) When the initial contact area is fixed, sharp microcones with larger height H experience smaller friction-induced energy dissipation, leading to high response-relaxation speed. However, sharp microcones may also undergo buckling instability that undermines mechanical stability.
- (4) Increasing the initial contact area lowers the sensitivity of the sensor, thus reducing the detectable pressure range.

Therefore, microcones in this work are conditionally optimal and simultaneously achieve low energy dissipation, high sensitivity, and high mechanical stability.

Fig. 2 Effect of microcone structure on response-relaxation time. **a**, Parameters of microcone's structure. **b**, Simulated energy loss versus pressure of microcones with various heights H . **c**, Contour plot of critical buckling pressure of 20 microcones. **d**, Simulated energy loss versus pressure of microcones with various bonded areas A . **e**, Normalized capacitance change versus pressure of microcones with various bonded areas. The dotted data are experimental results. **f**, Contour plot of

normalized energy loss of 20 microcones. **g**, SEM of microcone arrays with different bonding ratios. **h**, Response-relaxation time of microcones arrays with different bonding ratios.

On Page 6, we add:

Effect of microcone structure on response-relaxation time

The bonded microcones play a key role in reducing the energy dissipation during the contact-separation process, thus increasing the response-relaxation speed. The effect of microcone structure on the response-relaxation time is further investigated by FEA. The structure of a single microcone in 2D configuration can be described by three parameters: height H , top surface A_0 (i.e., initial contact area), and bottom surface D (Fig. 2a). 20 different microcones are investigated by varying $A_0/D=0.2, 0.4, 0.6, 0.8, 1$ and $H/D = 0.5, 1, 1.5, 2$. With a fixed $A_0/D = 0.4$, we first study the effect of height H on the energy dissipation by varying H/D . Results in Fig. 2b show that sharper microcones (i.e., larger H/D) show less energy loss. However, sharp microcones may also undergo buckling instability that undermines the mechanical stability of the sensor. For a tapered column with clamped top/bottom surfaces, the critical buckling force can be calculated by $\frac{4\pi^2 EI_D}{H^2} \frac{A_0^2}{D^2}$ where E is the Young's modulus and $I_D = \frac{\pi D^4}{64}$.⁵⁰ The critical buckling pressure (denoted as P_c) of 20 microcones are summarized in Fig. 2c. Considering that the maximum applied pressure should be greater than 300 kPa, microcones with small A_0/D and large H/D should not be adopted (marked with blue in Fig. 2c). Second, we analyze how initial contact area A_0 affect the energy dissipation by setting $H/D = 1$ and varying A_0/D . The normalized energy loss in Fig. 2d suggests that larger A_0/D yields a lower energy loss. This conclusion is expected because stout microcones (i.e., larger A_0/D) create less contact area under the same pressure while preserving more elastic energy W_{elastic} . However, we may not rush to conclude that microcone with $A_0/D = 1$ (i.e., a cylindrical micropillar) is the best by far, because the flat dielectric is notorious for its low sensitivity,²⁰ i.e., the slope of $\Delta C/C_0$ versus pressure curve. As also revealed by our models in Fig. 2e, we show that the sensitivity gradually drops when A_0/D increases (Supplementary Text 2), leading to an undermined detection range due to the saturated signals, as validated by experiments (dotted data in Fig. 2e). Therefore, the tradeoff between response-relaxation speed and sensitivity should also be considered. By summarizing the normalized energy loss of 20 microcones in Fig. 2f with background colors indicating other metrics, we show that microcones with moderate values of $(A_0/D, H/D)=(0.4, 1), (0.6, 1)$ and $(0.6, 1.5)$ can simultaneously achieve low energy dissipation, high sensitivity, and high mechanical stability. Note that it is difficult to fabricate bonded microcones with perfect geometries, and microcones used in this work have close values of $(A_0/D, H/D) \approx (0.4, 1)$ (see SEM in Fig. 1b). In addition to the structure of microcones, the bonding ratio of microcone arrays also affect the response-relaxation time of the sensor. In this regard, we compare the performance of six sensors with bonding ratios of 0, 20, 40, 60, 80, and 100% (Fig. 2g). Results in Fig. 2h manifest that the 100% bonded ratio provides the lowest response-relaxation time of ~ 0.04 ms, which is higher than previous sensors without fully bonded interfaces (Supplementary Fig. 5).²⁷

Comment #2

The authors claimed that the 0.1 ms response and relaxation time in Fig. 1 is the limitation of the test system, but not of the sensor. However, the test system shown in Supplementary Fig. 2 and 3 does not indicate any limitations.

- (1) Is the ADC sampling rate of the MCU the limiting factor? The reviewer would suggest the authors clarify the true limiting factor rather than simply provide schematics of the circuits.
- (2) If the circuit system is the bottleneck, the authors are suggested to use a better system to further investigate the true speed of the pressure sensor.

Response #2

Thanks for the suggestion. To investigate the true speed of the pressure sensor, we have updated the circuit of the customized testing system, increasing the sampling frequency bandwidth from 10 kHz to 25 kHz. The detailed design of the circuit board is now presented in **Supplementary Fig. 3** and measured data are correspondingly modified in the main text. Brief answers to your questions are as follows:

- (1) In fact, the ADC sampling rate of the MCU is not the limiting factor. The design principle of the circuit in our work involves measuring the charging time of the sensor under a specific DC voltage and resistance to calculate the capacitance value. The capacitance measurement period of the test circuit $t = t_{\text{charging}} + t_{\text{discharging}} + t_{\text{computation and communication}}$. Therefore, the limiting factors affecting the sampling frequency of the circuit primarily include the capacitance charging time t_{charging} , discharging time $t_{\text{discharging}}$ and charging time measurement accuracy. The charging time t_{charging} for the capacitor to charge to a certain voltage U_0 is determined by DC power supply U_E , series resistance R and the measured capacitor C . In order to ensure precise testing of our circuit within the capacitance range of 0-8000 pF, both the charging resistance and parallel capacitance are chosen to have relatively large values. According to the actual test results, t_{charging} should be 70 μs to avoid additional errors caused by too fast charging rates. In terms of capacitor discharge time, it is necessary to ensure that the voltage across the capacitor is discharged to zero in order to avoid interference with the next capacitor measurement and cumulative errors. Therefore, the discharging time is controlled above 20 μs . Considering the margin for other limiting factors such as data processing and switching time (10 μs), the measurement time is about 100 μs , resulting in a measurement frequency bandwidth of 10 kHz.
- (2) We have optimized the circuit and increased the sampling frequency to 25 kHz. The detailed modifications of the circuit are as follows: i. The resistance R_4 connected to pin 3 (OUT) of the precision timer SE555 was adjusted from 51 to 10 Ω to accommodate the measured pulse waveform under high frequency, thereby shortening the rise time of the high level. ii. The capacitance C_2 connected to pin 7 (Cap+) of the SE555 was reduced from 100 pF to 10 pF to align with software modifications, controlling the charging time of the tested capacitor within 25 μs . iii. The range-switching module of the original circuit was eliminated, and a fixed range connected to the tested capacitor with the shortest PCB trace path was implemented. The primary objective is

to reduce the lead length of the capacitor measurement circuit, minimize external interference noise picked up along the high-frequency analog signal traces, and enhance the accuracy and stability of capacitor measurements. iv. The triggering interval of the SE555 timer was reduced from 100 to 40 μs , and the trigger pulse width was decreased from 20 to 12 μs . Simultaneously, the charging time of the tested capacitor was further shortened to within 25 μs , allowing an increase in the sampling frequency bandwidth from 10 to 25 kHz. Additionally, precise equipment such as a signal generator and high-speed oscilloscope were employed to accurately measure the delay t_d between the trigger pulse and the beginning of charging. Compensation and calibration for this delay were achieved through software algorithms to mitigate additional measurement errors arising from the increased sampling frequency.

Furthermore, we also tried to use other principles to design the circuit. The AC-based capacitance measuring circuit has been verified to be used in high-precision and high-frequency capacitance measurement systems, and its design schematic is shown in Fig. R1. A sinusoidal voltage, V_{in} , is applied to the circuit input. Then the current flowing through the capacitor is transformed into voltage V_o by an operational amplifier. The capacitance being measured can be calculated by the output voltage V_o , $V_o = -V_{in} \frac{j\omega R_f C_x}{j\omega R_f C_f + 1}$. Then the capacitance should be modulated from the output voltage V_o . However, through practical testing, it was observed that capacitance signal cannot be identified under high-frequency conditions. Although the sampling frequency bandwidth can reach 100 kHz, the capacitance measurement accuracy is sacrificed too much, resulting in the inability to identify capacitance changes at high frequencies.

Fig. R1. Schematic diagram of AC-based capacitance measuring circuit.

Therefore, to simultaneously achieve a wide sampling frequency bandwidth and high capacitance testing accuracy, the maximum sampling frequency of our circuit is set at 25 kHz. Although we can only observe signals up to 12,500 Hz, which is the upper limit recognizable by the sensor, we believe that this does not represent the true response speed of our sensor. Given sufficient time, future endeavors may explore the design of circuit boards with more advanced technology to achieve a broader sampling frequency bandwidth and further investigate the actual response speed of the sensor.

Supplementary Fig. 3. Schematic design diagram of the digital circuit board. a, Schematic diagram of capacitor charging and discharging circuit and the correspondence between capacitance value and charging time. b, The capacitance measurement circuit that is mainly composed of five modules: power supply, range switching, capacitance measurement, microcontroller, and high-speed communication. c, Schematic diagram of capacitor charging and discharging circuit and the correspondence between capacitance value and charging time.

Comment #3

The sampling rate of the system appears to be varying during the author's experiments. In Fig. 1f, Fig. 2f, and Fig. 3b,g, the sampling frequencies appear to be different, causing different levels of details revealed in those figures. Please state clearly the experiment setup in each of the cases, and other cases mentioned in the paper if not listed here.

Response #3

Yes, the sampling rates are different. The sampling rates for Fig. 1f and Fig. 3b,g are the same, while that for Fig. 2f is different. The latter is measured using a commercial LCR meter rather than the home-made circuit. We have stated clearly the experiment setup in each of the measurements in the revised manuscript per the suggestion of the reviewer.

On page 15, we add:

The capacitance response was measured using an LCR meter (E4980AL, KEYSIGHT) at a testing frequency of 1 MHz if not specified. The loading and unloading cycles were conducted at a loading rate of $100 \text{ kPa}\cdot\text{s}^{-1}$ The capacitance responses for response time, relaxation time and mechanical vibration were measured using a home-made circuit with a sampling frequency of 25 kHz.

Comment #4

The author claimed that the reported pressure sensor could detect high-frequency responses. However, there are other flexible materials or flexible sensor structures that can detect MHz-level vibrations, such as PVDF and micromachined ultrasound transducers (DOI: 10.1038/s41378-018-0022-5). The advantage of the reported pressure sensor over these ultrasonic transducers remains unclear.

Response #4

Because of different sensing mechanisms, some other sensors, including PVDF-based sensors and the micromachined ultrasound transducers, have a much wider frequency bandwidth. By contrast, our sensor can respond to both static and dynamic pressures simultaneously. It is highly soft and stretchable, and it does not require a vacuum environment.

To alleviate your concerns, on Page 12, we add:

“Our sensor design represents a significant advancement over existing PVDF-based flexible sensors, which may possess a broader frequency bandwidth.⁵² Notably, our sensors can detect both static and dynamic pressures simultaneously. They are also softer and more flexible and do not require a vacuum environment, broadening their practical applications in fields such as robotics, the metaverse, and biomedical engineering.”

Comment #5

The artificial ear sensor has a resonance frequency of 200 Hz, and the response to high-frequency vibrations (>1000 Hz) is ~4 times weaker. This may compromise the sound recording fidelity. Can the pressure sensor, eardrum or ear canal be further engineered to optimize the sensor response to the high-frequency band?

Response #5

Great point! Yes, the amplitude of the signal at high frequencies may be improved by further engineering the sensor, the circuit, and the eardrum structure. For example, the signal amplitude at 1000 Hz is enhanced by using a smaller eardrum thickness (Fig R2a), or by reducing the mass of the ossicle (Fig R2b). However, there is still a big difference in signal magnitude at different frequencies, and thus a calibration may be required to substantially improve the recording fidelity. However, all those are engineering issues and are out of the scope of this work, although we are working on those to make a better system.

Fig R2. a, The capacitive response of the eardrum with the thickness of 25, 50, and 100 μm at a high frequency of 1000 Hz. b, The capacitive response of the ossicle with the mass of 0.5, 1.1, and 2.3 g at a high frequency of 1000 Hz.

Comment #6

The authors used carbon nanotubes as the doping material. The reviewer would appreciate a brief justification of the preference for CNTs over other nanomaterials that may obtain similar functions.

Response #6

Thank you for your suggestion. A brief justification is added on Page 8:

“The choice of CNTs is due to their stable mechanical and superior electrical properties. Notably, the PDMS-CNTs composite’s relative permittivity is significantly higher than that of pure PDMS, thereby enhancing the capacitance signal.⁵¹”

Minor:

On Page 6, Line 112, the unit ‘ms’ should be removed from the equation to ensure the unit of f_{max} is Hz.

Response

Done as suggested.

Reviewer #3

General comments

This manuscript presents an ultrafast piezocapacitive pressure sensor with a bonded microstructure. Compared to conventional piezocapacitive type pressure sensor, this one exhibits a rapid response time of 0.1 ms, in contrast to the conventional tens of ms. This performance was achieved using a bonded, microstructured interface, which reduces interfacial elastic instability and viscoelastic dissipation. Due to its high response and relaxation speeds, it can detect not only dynamic forces but is also suitable for acoustic applications. However, the novelty and innovation of this article are not sufficient for publication in Nature Communications. A similar sensor structure and working mechanism have already been reported in Nature Communications by the same authors. There was only a minor difference compared to the previous publication (Nat. Commun. 2022, 13, 1317) Therefore, I do not recommend the publication of this manuscript in Nature Communications.

Response: We would like to thank the referee for reviewing the manuscript and providing useful suggestions to further improve our work. We need to clarify that this work is substantially different from our previous paper (Nat. Commun. 2022, 13, 1317).

- (1) The two works focus on different topics. This work focuses on the response-relaxation speed of sensors, while the previous work focuses on mechanical stability.
- (2) The two works provide substantially different scientific understanding. This paper provides a deep understanding of the response-relaxation speed that has not been reported before. Based on our understanding, we can increase the frequency bandwidth from 100 Hz level to ~10 kHz, which is a **huge improvement** compared to previous works. By contrast, our published paper (Nat. Commun. 2022, 13, 1317) provides a structure-induced toughening mechanism.
- (3) The two works use different sensor structures, although the difference is not big. In our published paper (Nat. Commun. 2022, 13, 1317), 20%~30% cones are not bonded (Supplementary Fig. xx). Such non-bonded cones will lead to a total response-relaxation time of ~1 ms, or a frequency bandwidth of ~1 kHz. In this work, however, all cones are bonded, the response-relaxation time decreases to less than 0.04 ms, and the corresponding frequency bandwidth significantly expands to 10 kHz. Although it seems that the difference is small, the change in result is big (~10 times).

Supplementary Fig. 5. a, SEM image of cross-section of the sensor in previous work (Nat. Commun. 2022, 13, 1317) with 70%~80% bonding ratio. b, SEM image of cross-section of the sensor in this work with 100% bonding ratio.

Comment #1

1. In the author's previous paper (Nat Commun. 2022, 13, 1317), the device closely resembles the one presented in this work. Even though both devices share the same material composition, structure, mechanism, and the idea of topological interlinks, they yield different results. The points of difference are as follows.

- (1) Both devices feature the same topological interlinks and structure, yet they differ in interfacial toughness.
- (2) In this manuscript, the authors achieve reduced response and relaxation times due to the topological interlinks. However, in the prior paper, despite the presence of these interlinks, the device exhibited a significantly longer response and relaxation time (~ 6 ms).
- (3) The device in this manuscript has a reduced sensitivity to the pressure compared to the one in the previous study.
- (4) The device in this manuscript has a lower detection limit for pressure than the one in the previous study.

Response #1

(1) Two structures have different interfacial bonding ratio that leads to different interfacial toughness. As shown in Supplementary Fig. xx, the microstructure in previous work was replicated from the *Calathea zebrine* leaves. The structures are not highly uniform, and part of the cones are not bonded. The bonding ratio is about 80%. In this work, the microcones are made by a 3D-printing routine. All cones are uniform in height, and they are all bonded. Therefore, they exhibit different interfacial toughness.

(2) First, as shown in the revised Fig. 2f, the sensor with a 80% bonding ratio shows a response-relaxation time of ~ 1 ms, while the fully bonded sensor shows significantly smaller response-relaxation time of 0.04 ms. Second, the longer response-relaxation time of ~ 6 ms in previous work was detected using a commercial LCR meter (E4980AL, KEYSIGHT) that has a detection limitation of 6 ms. In this work, we customize a circuit board that can detect high frequency up to 25 kHz, i.e., 0.04 ms.

Fig. 2. g, SEM of microcone arrays with different bonding ratios. **h**, Response-relaxation time of microcones arrays with different bonding ratios.

(3) We agree with the reviewer that the magnitude of the signal in this work is about 50% of the previous one. The decrease in sensitivity is attributed to the larger initial capacitance of the pressure sensor here, i.e., C_0 . In the previous work, some microstructures were not bonded, resulting in a smaller initial capacitance.

(4) The inconsistency in detection limits arises from differences in the testing methods. The detection limit in this work is tested using acoustic pressure. In contrast, the previous work used a force gauge with a computer-controlled stage to test the detection limit.

To alleviate the concerns of the reviewer, we added the following descriptions in the revised manuscript.

On Page 7, we add:

“In addition to the structure of microcones, the bonding ratio of microcone arrays also affect the response-relaxation time of the sensor. In this regard, we compare the performance of six sensors with bonding ratios of 0, 20, 40, 60, 80, and 100% (Fig. 2g). Results in Fig. 2h manifest that the 100% bonded ratio provides the lowest response-relaxation time of ~ 0.04 ms, which is higher than previous sensors without fully bonded interfaces (Supplementary Fig. 5).²⁷”

Comment #2

In Fig. 4d, the device displays a resonant frequency at 200 Hz and exhibits a nonlinear response up to 5000 Hz. However, in Fig. 3d, 3f, 3g, the signal maintains the same amplitude across different frequencies. Why doesn't the signal exhibit frequency dependence.

Response #2

Many thanks for pointing out the difference. In fact, the change in signal magnitude at different frequencies reflects the property of the artificial tympanic membrane instead of the sensor. The resonant frequency of the artificial tympanic membrane is 200 Hz, which is unrelated to the performance of the sensor. By contrast, in Fig. 3d, 3f, 3g, the vibration was generated by a vibration generator, and we tried to align the amplitudes of different vibrations, and thus the magnitudes of the signals at different frequencies are close.

Comment #3

In Fig. 4a, there's a depiction of the sensor output signal processing using deep learning for phrase reconstruction. However, within the main content of this paper, only some STFT data is provided for signal processing, and there is no mention or elaboration on signal processing and reconstruction through deep learning as indicated in the figure. Consequently, the section of the figure pertaining to signal processing and reconstruction should be removed.

Response #3

We are sorry for the confusion. In fact, we constructed a deep learning model but the data were not used in this paper. Now we have removed the content of deep learning in Fig. 5a. Sorry again for the carelessness.

Fig. 5. a, Schematics of the biological system and the artificial system for sound detection.

Comment #3

In the study, the author employed 2 wt% and 7 wt% CNT/PDMS composites for the dielectric layer and electrode layer, respectively. To bolster the integrity and rationale of this work, it would be beneficial if the author provided justifications for selecting these specific concentrations. For instance, data or references on electrical conductance, dielectric constant, and dielectric loss could substantiate the author's choice of material composition.

Response #4

We thank the reviewer for the suggestion. The selection of materials is based on our previous study (*Nat Commun.* 2022, 13, 1317), so that we have cited the paper to substantiate our selection per the suggestion of the reviewer.

We use 2 wt% CNT-PDMS composite as the dielectric because it is close to the percolation threshold. This composition can have a large pressure dependent permittivity and thereby high sensitivity. But if the concentration goes higher, the composite becomes a conductor. The reason that we select the composite with 7 wt% CNTs for electrodes is that this composition has an acceptable balance of electrical conductance and mechanical properties.

Comment #5

It is recommended to add data regarding the characteristics of the microstructure, specifically related

to its design, density, or the extent of the interlinked area

Response #5

Thanks for the suggestion. The design of the microstructure in this work is conditionally optimal and simultaneously achieves low energy dissipation, high sensitivity, and high mechanical stability. To elucidate the microstructure effects on the performance of the sensor, we have carried out both theoretical modeling and finite element analyses of 20 different microcones. Results are now presented in the newly added **Fig. 2** and the main text “**Effect of microcone structure on response-relaxation time**”.

On Page 6, we add:

Effect of microcone structure on response-relaxation time

The bonded microcones play a key role in reducing the energy dissipation during the contact-separation process, thus increasing the response-relaxation speed. The effect of microcone structure on the response-relaxation time is further investigated by FEA. The structure of a single microcone in 2D configuration can be described by three parameters: height H , top surface A_0 (i.e., initial contact area), and bottom surface D (Fig. 2a). 20 different microcones are investigated by varying $A_0/D=0.2, 0.4, 0.6, 0.8, 1$ and $H/D = 0.5, 1, 1.5, 2$. With a fixed $A_0/D = 0.4$, we first study the effect of height H on the energy dissipation by varying H/D . Results in Fig. 2b show that sharper microcones (i.e., larger H/D) show less energy loss. However, sharp microcones may also undergo buckling instability that undermines the mechanical stability of the sensor. For a tapered column with clamped top/bottom surfaces, the critical buckling force can be calculated by $\frac{4\pi^2 EI_D}{H^2} \frac{A_0^2}{D^2}$ where E is the Young’s modulus and

$I_D = \frac{\pi D^4}{64}$.⁵⁰ The critical buckling pressure (denoted as P_c) of 20 microcones are summarized in Fig. 2c. Considering that the maximum applied pressure should be greater than 300 kPa, microcones with small A_0/D and large H/D should not be adopted (marked with blue in Fig. 2c). Second, we analyze how initial contact area A_0 affect the energy dissipation by setting $H/D = 1$ and varying A_0/D . The normalized energy loss in Fig. 2d suggests that larger A_0/D yields a lower energy loss. This conclusion is expected because stout microcones (i.e., larger A_0/D) create less contact area under the same pressure while preserving more elastic energy W_{elastic} . However, we may not rush to conclude that microcone with $A_0/D = 1$ (i.e., a cylindrical micropillar) is the best by far, because the flat dielectric is notorious for its low sensitivity,²⁰ i.e., the slope of $\Delta C/C_0$ versus pressure curve. As also revealed by our models in Fig. 2e, we show that the sensitivity gradually drops when A_0/D increases (Supplementary Text 2), leading to an undermined detection range due to the saturated signals, as validated by experiments (dotted data in Fig. 2e). Therefore, the tradeoff between response-relaxation speed and sensitivity should also be considered. By summarizing the normalized energy loss of 20 microcones in Fig. 2f with background colors indicating other metrics, we show that microcones with moderate values of $(A_0/D, H/D)=(0.4, 1), (0.6, 1)$ and $(0.6, 1.5)$ can simultaneously achieve low energy dissipation, high sensitivity, and high mechanical stability. Note that it is difficult to fabricate bonded microcones with perfect geometries, and microcones used in this work have close values of $(A_0/D, H/D) \approx (0.4, 1)$ (see SEM in Fig. 1b). In addition to the structure of microcones, the bonding ratio of microcone arrays also affect the response-relaxation time of the sensor. In this regard, we compare the performance of six

sensors with bonding ratios of 0, 20, 40, 60, 80, and 100% (Fig. 2g). Results in Fig. 2h manifest that the 100% bonded ratio provides the lowest response-relaxation time of ~ 0.04 ms, which is higher than previous sensors without fully bonded interfaces (Supplementary Fig. 5).²⁷

Fig. 2 Effect of microcone structure on response-relaxation time. **a**, Parameters of microcone's structure. **b**, Simulated energy loss versus pressure of microcones with various heights H . **c**, Contour plot of critical buckling pressure of 20 microcones. **d**, Simulated energy loss versus pressure of microcones with various bonded areas A . **e**, Normalized capacitance change versus pressure of microcones with various bonded areas. The dotted data are experimental results. **f**, Contour plot of normalized energy loss of 20 microcones. **g**, SEM of microcone arrays with different bonding ratios. **h**, Response-relaxation time of microcones arrays with different bonding ratios.

On Page 14, we add:

A resin template with a microcone array with cone diameter of $50 \mu\text{m}$ and height of $40 \mu\text{m}$ was fabricated using high precision 3D printing (NanoArch S130, BMF Precision Tech, Inc.). The density of the microcones is $4 \times 10^4 \text{ cm}^{-2}$. The bonding joint is about $2 \mu\text{m}$ in diameter.

Comment #6

On page 9, the text states, "Our sensor also identifies high-frequency vibration under a pre-applied static pressure or low-frequency stimuli." How does the sensor perform when there is no pre-applied static pressure or low-frequency stimuli?

Response #6

Unless a preload is specified, the response-relaxation time of our sensor is tested by a vibration generator (Model BL-ZDQ-2185, Hangzhou Peilin Instrument Co. Ltd.) without a preload. Results without preloads are presented in Fig. 3d and 3e. To alleviate your concern, we clearly state this in **Methods** on Page 15:

“Unless a preload is specified, a vibration generator (Model BL-ZDQ-2185, Hangzhou Peilin Instrument Co. Ltd.) was adopted to apply vibration of given frequencies to the sensor.”

REVIEWER COMMENTS

Reviewer #1 (Remarks to the Author):

The authors' responses have successfully answered my comments. This paper can be accepted by *Natura Communications*.

Reviewer #2 (Remarks to the Author):

The authors have made significant improvements to the manuscript by revising the text and supplementing additional experimental results, which substantiate their claims more robustly. The manuscript's quality is much improved, while the reviewer reserves minor concerns regarding technical accuracy within the discussions. To further improve the manuscript quality, the reviewer recommends that the authors address the following issues:

1. The authors admit that their sensor has big frequency-response variations in their answers to comment#5. To address this issue, the manuscript could benefit from additional discussions on potential solutions, such as integrating multiple eardrums with varying thicknesses or incorporating frequency-dependent gain adjustments to compensate for the observed variations in frequency response. Such enhancements could significantly improve the manuscript by demonstrating the practical feasibility of using this novel sensor technology in artificial ear applications.
2. In addition, as the sensor has significant response variations in frequency (the response to 1000 Hz is ~4 times weaker than that at 200 Hz), it is confusing why the sensor with a bonded interface has almost the same recordings as the cellphone microphone when sensing a wide-band signal (0-1 kHz) as shown in Fig. 5e and f. The frequency response and recording fidelity are as good as a wideband microphone, which is too good to be true.
3. On Page 12, Lines 260-261, the author discusses the limitation of the micromachined ultrasound transducers, specifically mentioning that these sensors require vacuum environments for operation, which purportedly limits their practical applications. However, the vacuum condition is only required during the fabrication process. In fact, acoustic (mechanical) waves do not propagate in a vacuum, they can operate in air, liquid, or solid tissue.
4. The FEM simulation depicted in Fig. 1c is greatly valued. Nonetheless, it appears that only the loading process is shown for the interface-bounded device. It's worth noting that the unloading process may not be reversible for such devices, as the bounding interface undergoes changes during loading. Hence, presenting the unloading process independently would be the optimal approach for comprehensively characterizing the system.

Reviewer #3 (Remarks to the Author):

Despite the authors' efforts to differentiate their current work from their previous publication, the incremental advances presented do not sufficiently meet the high novelty threshold expected by Nature Communications. The enhancement in sensor response-relaxation speed, while technically significant, appears to build incrementally on the previous material system (bonded microstructure) rather than introducing a groundbreaking scientific breakthrough or an innovative material or sensor structure. Regarding fast response times, I found several papers (Liu et al., ACS Appl. Mater. Interfaces 2017, 9, 24148–24154; Lee et al., ACS Nano 2021, 15, 1795–1804; Xu et al., ACS Appl. Mater. Interfaces 2021, 13, 38792–38798; etc.) that reported frequency bandwidths between 10-20 kHz based on capacitive and resistive sensors. Among piezoelectric or triboelectric sensors, numerous publications demonstrate bandwidths over 10 kHz. Therefore, I believe this work does not demonstrate significant advancements compared to the authors' previous publication.

Response to reviewers for manuscript NCOMMS-23-38129A

We thank all reviewers for your constructive comments and suggestions to improve our work. In this letter, your comments are addressed carefully in a point-by-point manner. For your convenience, the modifications in the revised manuscript are marked in blue. The point-by-point responses are listed below.

Reviewer #1:

The authors' responses have successfully answered my comments. This paper can be accepted by Natura Communications.

Response: We thank the reviewer for suggesting publication of our work in Nature Communications.

Reviewer #2:

General comments:

The authors have made significant improvements to the manuscript by revising the text and supplementing additional experimental results, which substantiate their claims more robustly. The manuscript's quality is much improved, while the reviewer reserves minor concerns regarding technical accuracy within the discussions. To further improve the manuscript quality, the reviewer recommends that the authors address the following issues:

Response: We sincerely thank the reviewer for the recognition of the quality improvement. The minor comments raised by the reviewer will definitely further strengthen our work.

Comment #1

The authors admit that their sensor has big frequency-response variations in their answers to comment#5. To address this issue, the manuscript could benefit from additional discussions on potential solutions, such as integrating multiple eardrums with varying thicknesses or incorporating frequency-dependent gain adjustments to compensate for the observed variations in frequency response. Such enhancements could significantly improve the manuscript by demonstrating the practical feasibility of using this novel sensor technology in artificial ear applications.

Response#1

Many thanks for the suggestion. The additional discussions have been added to the revised manuscript. On Page 12, we add:

Although our artificial system can effectively showcase the practical feasibility of using this sensor in artificial ear applications, there is a large frequency-response variation due to the single structure. This frequency-response behavior can be optimized by integrating multiple tympanic membranes of different thicknesses or by incorporating frequency-dependent gain adjustments to compensate for the frequency-response variations.

Comment #2

In addition, as the sensor has significant response variations in frequency (the response to 1000 Hz is ~ 4 times weaker than that at 200 Hz), it is confusing why the sensor with a bonded interface has almost the same recordings as the cellphone microphone when sensing a wide-band signal (0-1 kHz) as shown in Fig. 5e and f. The frequency response and recording fidelity are as good as a wideband microphone, which is too good to be true.

Response #2

We thank the reviewer for pointing out the problem. Yes, the recordings for the two cases should be somehow different. The similar recordings shown in Fig. 5e and f are because we performed filtering and noise reduction processing, which remove the fine structures of the waveforms. Another reason for the similarity lies in that the selected music frequencies are above 200 Hz. A closer look at Figure 5d reveals a relatively flat region after 200 Hz. Consequently, the difference between the time domain signals identified by the microphone and the sensor is not striking.

Now, we use the raw data without performing any data filtering or noise reduction. The updated results are presented in Fig. 5e and f, which show small differences between the frequency response of the cellphone microphone and that of our sensor.

Fig. 5 e, Acoustic signal waveforms including waveforms recorded by a cellphone, waveforms converted from the sensor with a bonded interface, and waveforms converted from the sensor with a non-bonded interface. **f**, Corresponding STFT spectrograms of the acquired sound waveforms in panel (e).

Comment 3

On Page 12, Lines 260-261, the author discusses the limitation of the micromachined ultrasound transducers, specifically mentioning that these sensors require vacuum environments for operation, which purportedly limits their practical applications. However, the vacuum condition is only required during the fabrication process. In fact, acoustic (mechanical) waves do not propagate in a vacuum, they can operate in air, liquid, or solid tissue.

Response #3

We thank the reviewer for pointing out our improper description. We now removed the description

regarding vacuum environment. On Page 13, the revised description reads: Notably, our sensors can detect both static and dynamic pressures simultaneously. They are also softer and more flexible, broadening their practical applications in fields such as robotics, the metaverse, and biomedical engineering.

Comment #4

The FEM simulation depicted in Fig. 1c is greatly valued. Nonetheless, it appears that only the loading process is shown for the interface-bounded device. It's worth noting that the unloading process may not be reversible for such devices, as the bounding interface undergoes changes during loading. Hence, presenting the unloading process independently would be the optimal approach for comprehensively characterizing the system.

Response #4

The reviewer is correct. As seen in Fig. 1d, the contact area during loading and unloading are not fully reversible under the same pressure. But their difference is not significant. In the revised manuscript, the simulated results for unloading process have been added as Supplementary Fig. 3. On Page 5, we add “The corresponding unloading process is shown in Supplementary Fig. 3.”

Supplementary Fig. 3. Finite element simulations of both non-bonded and bonded pressure sensors under the unloading process.

Reviewer #3:

General comments:

Despite the authors' efforts to differentiate their current work from their previous publication, the incremental advances presented do not sufficiently meet the high novelty threshold expected by Nature Communications. The enhancement in sensor response-relaxation speed, while technically significant, appears to build incrementally on the previous material system (bonded microstructure) rather than introducing a groundbreaking scientific breakthrough or an innovative material or sensor structure. Regarding fast response times, I found several papers (Liu et al., ACS Appl. Mater. Interfaces 2017, 9, 24148–24154; Lee et al., ACS Nano 2021, 15, 1795–1804; Xu et al., ACS Appl. Mater. Interfaces 2021, 13, 38792–38798; etc.) that reported frequency bandwidths between 10-20 kHz based on capacitive and resistive sensors. Among piezoelectric or triboelectric sensors, numerous publications demonstrate bandwidths over 10 kHz. Therefore, I believe this work does not demonstrate significant advancements compared to the authors' previous publication.

Response:

We appreciate the reviewer for the comments and the opportunity to further clarify the novelty and significance of our capacitive soft pressure sensor. Unlike existing resistive-based or traditional hard Si-based sensors, our work not only introduces a unique capacitive approach but also is the first to elucidate the mechanism of enhanced response-relaxation speed through detailed modeling and simulation. The new understanding also leads to new results: we extend the frequency range of capacitive sensors from 100 Hz level to 10 kHz level.

While the frequency bandwidths between 10-20 kHz have been reported, such as the previous works by Liu et al. (ACS Appl. Mater. Interfaces 2017, 9, 24148–24154) and Lee et al. (ACS Nano 2021, 15, 1795–1804), these studies focus on resistive pressure sensors. Our capacitive sensor operates on a fundamentally different principle, offering distinct advantages.

The other work by Xu et al. (ACS Appl. Mater. Interfaces 2021, 13, 38792–38798) focuses on a traditional Si-based sensor, which is hard and brittle. By contrast, our sensor is fully soft and stretchable. The soft sensors can have a much wider range of applications that may not be realized using traditional hard sensors.

Our sensor's ability to detect both static and high-frequency dynamic signals over 10 kHz represents a significant leap over existing technologies, which typically excel in only one area. Piezoelectric and triboelectric sensors, for instance, are limited to dynamic signal detection and cannot capture static signals. Our sensor, however, can not only detect static pressure but also record high-frequency vibrations up to 10 kHz, marking a notable advancement in sensor technology.

In summary, our capacitive soft pressure sensor not only diverges fundamentally from existing resistive and Si-based sensors in design and operation but also sets a new benchmark in the field through its dual capability of detecting both static and high-frequency dynamic signals, bolstered by our pioneering research into its underlying mechanisms. This represents a substantial forward leap in sensor technology.

On Page 13, we add the following sentences in Discussion: Note that there are sensors that have similar or wider bandwidth ranges, including traditional silicon-based capacitive sensors,⁵³ and flexible piezoelectric and triboelectric sensors.^{54,55} However, the traditional silicon-based capacitive sensors are stiff, while the piezoelectric and triboelectric sensors are limited to the detection of dynamic signals. By contrast, our capacitive sensor is soft and has little signal drift;⁵⁶ it can not only detect static pressure

but also record high-frequency vibrations over 10 kHz, making it potential for a wider range of applications.